# Transformer-based Unsupervised Graph Domain Adaptation

## Abstract

Unsupervised Graph Domain Adaptation (UGDA) addresses the challenge of domain shift in transferring knowledge from a labeled source graph to an unlabeled target graph. The existing UGDA methods are based solely on graph neural networks (GNNs) with limited receptive fields. This characterization of UGDA methods constrains their ability to capture long-range dependencies between domains and effectively adapt to sparse graph domains. To overcome this limitation, we introduce a novel transformer-based UGDA (TUGDA) framework that sequentially integrates transformers and asymmetric GCNs to capture both global and local structural dependencies between source and target graphs. Our framework leverages a transformer backbone, enriched with centrality and spatial positional encodings to enhance structural information. We further propose a new cross-attention mechanism that explicitly aligns source and target representations, along with theoretical analysis for reducing domain divergence through Wasserstein distance minimization. Extensive experiments on six cross-domain tasks in three real-world homophilic citation graphs show significant improvements over SOTA UGDA baselines. Our results validate TUGDA's ability to learn transferable, domain-invariant representations. Critically, to address practical scenarios where privacy or security constraints restrict access to real source domains, we demonstrate that TUGDA maintains strong performance using synthetic source graphs generated by a foundational model, outperforming leading baselines by up to 15% across six cross-domain tasks.

## 1 Introduction

Graph Neural Networks (GNNs) have shown remarkable success in various applications, including social network analysis Hamilton et al. (2017), traffic analysis Yu et al. (2018); Neshatfar et al. (2023), biomedical cell analysis Neshatfar & Sekeh (2024) and protein-protein interaction modeling Veličković et al. (2018). However, their reliance on extensive labeled graph data remains a critical bottleneck due to high annotation costs and domain-specific expertise Wu et al. (2020b).

To mitigate this challenge, Unsupervised Graph Domain Adaptation (UGDA) has emerged as a promising area of research in graph learning, including node classification problems. UGDA aims to transfer knowledge from a labeled source graph to an unlabeled target graph with the advantage of robustness against distributional shift between source and target domains Shen et al. (2020c); Liu et al. (2024a). Related research can be categorized as follows: **Foundational Graph Representation Methods**, includes self-supervised and unsupervised approaches that learn meaningful graph representations ( Wu et al. (2020a; 2023a); Liu et al. (2023b; 2024b)). **Early Graph Domain Adaptation Approaches** adapt established domain adaptation techniques without fully exploiting graph-specific structural properties, including adversarial alignment Shen et al. (2020c); Dai et al. (2022), statistical feature alignment Shen et al. (2020b), and structural adaptations You et al. (2023). Notably, **Advanced Graph-Specific Domain Adaptation Models** incorporate graph-specific mechanisms to preserve structural and semantic consistency across domains. These models emphasize structural adaptation, such as smoothing techniques for structural coherence Chen et al. (2025), and asymmetric propagation strategies demonstrated by A2GNN Liu et al. (2024a). Recent studies have also explored more advanced frameworks such as unfolded graph neural networks Zhang & Fink (2025), spectral domain adaptation with disentangled representations Yang et al. (2025), and

attribute-driven alignment (Fang et al., 2025a), highlighting the critical role of model architecture and spectral alignment in addressing structural and feature distribution shifts.

Despite these advances, most existing UGDA approaches suffer from a critical limitation: they rely heavily on conventional GNN architectures, which may struggle to model long-range dependencies and global structural contexts. This locality bias is particularly problematic in sparse graphs, such as citation networks, where distant nodes may have important semantic relationships that traditional GNNs miss due to their limited receptive fields. For example, distant papers in the same research area may never be reached by k-layer GNNs, despite having strong semantic relationships.

Attention-based architectures, particularly graph transformers, have emerged as powerful alternatives that enable capturing global contextual relationships and modeling complex long-range dependencies Ying et al. (2021); Zhuo et al. (2025); Liu et al. (2023a). By leveraging self-attention mechanisms to explicitly model pairwise interactions among nodes, transformers facilitate effective representation of global structural patterns across distant graph regions while incorporating graph-specific positional encodings for topological information integration.

Inspired by graph transformers, we propose *(TUGDA)*, a novel framework that addresses a fundamental limitation of existing UGDA methods: topology-constrained alignment. While methods like A2GNN achieve domain adaptation through structure-mediated message passing, requiring semantic correspondences to be topologically reachable, our core novelty introduces complementary alignment channels: (1) topology-independent semantic alignment via cross-attention that learns feature-space correspondences unconstrained by graph edges, (2) sequential transformer-GCN integration, where global semantic alignment precedes structural refinement, avoiding the bottleneck of topology-mediated adaptation, and (3) dual-alignment strategies combining explicit cross-domain attention with distributional regularization. This architectural design enables TUGDA to significantly outperform UGDA baselines across multiple datasets (see Table 1), particularly under the structural domain shifts. TUGDA integrates self-attention and cross-attention mechanisms to capture both intra-domain contexts and inter-domain semantic correspondences independent of topology. We enhance the transformer backbone with graph-specific positional encodings inspired by Ying et al. (2021), while introducing architectural modifications including sequential rather than parallel transformer-GCN composition and reformulated attention computation to handle source and target domains with different node dimensions.

The transformer backbone of TUGDA leverages SGFormer Wu et al. (2023b) for its efficient global attention computation with linear complexity, enabling scalability to large graphs. Unlike other transformers, SGFormer facilitates a more practical cross-attention implementation due to its flexible attention computation ordering, allowing for efficient interaction between source and target domain representations with different node sizes.

Our architecture takes a sequential approach: The transformers first derive global attention-based representations, which are then refined by asymmetric GCN propagation strategies similar to the A2GNN model. Our theoretical analysis reveals that the cross-attention mechanism inherently minimizes domain discrepancies by explicitly aligning source and target feature spaces, significantly reducing the Wasserstein divergence between domains. The subsequent GCN propagation enforces local smoothness, acting as a low-pass filter that reduces domain-specific noise and promotes the learning of transferable representations.

Extensive experiments against leading UGDA baselines demonstrate that TUGDA consistently outperforms existing methods, achieving substantial improvements in node classification across diverse graph adaptation scenarios.

In practice, UGDA faces an additional challenge: access to source domain data may be restricted due to privacy regulations, security concerns, or proprietary constraints. Healthcare networks, financial transaction graphs, and corporate communication networks often cannot be shared directly, necessitating synthetic alternatives that preserve statistical properties while protecting sensitive information. This scenario demands domain adaptation methods that can effectively transfer knowledge from synthetically generated source graphs to real target domains. Using synthetic graphs from GraphMaker Li et al. (2023), we show that TUGDA significantly outperforms A2GNN, demonstrating strong generalizability to privacy- and security-constrained settings.

**Our main contributions are as follows:**

- We introduce TUGDA, featuring a novel dual-alignment mechanism that combines topology-independent semantic alignment via cross-attention with structure-aware refinement through asymmetric GCN propagation.

- We propose a dual-alignment mechanism combining cross-attention for feature alignment and regularization loss for distributional matching.

- We demonstrate superior performance over state-of-the-art UGDA methods and show robust generalization to synthetic source domains generated by a foundational model.

## 2 PRELIMINARIES

### 2.1 PROBLEM DEFINITION

Given two graph domains source (s) and target (t), a labeled source graph $G^s = (\mathcal{V}^s, \mathcal{E}^s, X^s, Y^s)$ with node size $N^s$ and an unlabeled target graph $G^t = (\mathcal{V}^t, \mathcal{E}^t, X^t)$ with node size $N^t$, UGDA learns the model $f : G^t \to Y^t$ that predict node labels accurately on the target domain. In this work, $X \in \mathbb{R}^{N \times F}$ represents node features with $F$ dimensions. UGDA typically assumes a covariate shift scenario. This means that the conditional distributions of node labels given the graph remain consistent across domains, $P(Y|G) = P(Y|G)$, while the marginal distributions of node features and graph structures differ, $P(G) \neq P(G)$. The goal is thus to transfer the learned knowledge from the source domain to the target domain effectively despite these distribution discrepancies.

## 3 METHODOLOGY

We propose a novel UGDA framework that sequentially integrates transformer architectures with graph convolutional networks (GCNs). Our approach uses dual attention mechanisms, self-attention and cross-attention, in the transformers to effectively capture both intra- and inter-domain dependencies. Our TUGDA framework builds on positional encodings (PEs) inspired by Graphormer Ying et al. (2021) and an efficient transformer backbone adapted from SGFormer Wu et al. (2023b). Our overall objective function aligns with A2GNN Liu et al. (2024a), which comprises source and target encoders that are optimized using classification and domain alignment losses. Figure 1 demonstrates the architecture of the TUGDA method. Additionally, the algorithm of this model is available in the Appendix.

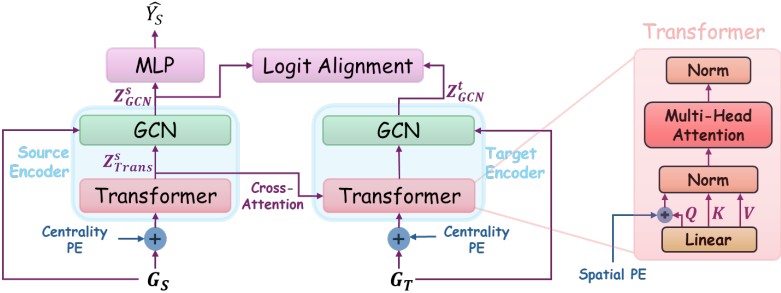

Figure 1: **TUGDA Architecture**: The source and target encoders share parameters and process their respective graphs, $G_S$ and $G_T$. The transformer modules generate query, key, and value representations, denoted as $Q$, $K$, $V$, and then applies the normalizations and matrix multiplications. $Z^s_{Trans}$ represents the transformer-encoded output from the source encoder. $Z^s_{GCN}$ and $Z^t_{GCN}$ are the final representations produced by the GCNs for the source and target domains, respectively.

### 3.1 POSITIONAL ENCODING IN PREPROCESSING

Positional encodings are essential in transformers to inject structural information into input sequences. To inject graph topology, we employ two key structural encodings from Graphormer Ying et al. (2021).

**Centrality Encoding:** Inspired by Graphormer, each node receives learnable embeddings based on its degree centrality, added into initial node features: $z_i^{(0)} = x_i + \xi_i(D)$. Here, $z_i^{(0)}$ is the model input vector corresponding to node $i$, and $\xi_i(.)$ is a learnable embedding function applied on the node degree vector $D$, where $D_i = \sum_j A_{ij}$.

## 3.2 Graph Transformer Backbone

Given node embeddings $z^{(0)} \in \mathbb{R}^{N \times F}$, the transformer computes query $Q$, key $K$, and value $V$ matrices as $Q = f_Q(z^{(0)})$, $K = f_K(z^{(0)})$, $V = f_V(z^{(0)})$, where $f_Q, f_K, f_V$ are linear projections.

**Spatial Positional Encoding:** To encode spatial relations, we adapt Graphormer's spatial encoding strategy to our transformer backbone. We add it to the normalized query, $\tilde{Q} = \frac{Q}{\|Q\|_F}$, instead of the attention logits, to consider the structure and nodes' distances in attention computation: $\hat{\tilde{Q}} = \tilde{Q} + b_{\phi(i,j)}$. Here, $\phi(i, j)$ denotes the shortest path distance (SPD) between nodes $i$ and $j$, and $b_{\phi(i,j)} \in \mathbb{R}^{N \times k}$ is a learnable presentation with $k$ being the transformer hidden dimension and $N$ is total number of nodes. The centrality and spatial positional encodings allow the Transformer to incorporate the graph-specific structure in the encoders.

**Global Attention:** The global attention output $Z_{\text{Trans}}$ is computed as:

$$Z_{\text{Trans}} = \beta D^{-1} \left[ \frac{1}{N} \hat{\tilde{Q}} (\tilde{K}^\top V) \right] + (1 - \beta) h^{(0)}, \tag{1}$$

where $\beta$ is linear combination multiplier for the residual link and is treated as a hyperparameter in our experiments. Further, $\tilde{K} = \frac{K}{\|K\|_F}$, $\| \cdot \|_F$ denotes the Frobenius norm, and $D = \text{diag}\left(1 + \frac{1}{N} \tilde{Q}(\tilde{K}^\top \mathbf{1})\right)$. This formulation enables efficient computation of attention-based message passing with linear complexity $O(N)$ relative to the number of nodes (Wu et al. (2023b)).

Beyond efficiency, this formulation has another advantage over transformers like Graphormer: the order of matrix multiplications enables cross-attention computation where $Q \in \mathbb{R}^{N^t \times k}$ and $V, K \in \mathbb{R}^{N^s \times k}$ are computed from the output of the source transformer. However, unlike SG-Former, which showed that a single-layer global attention can perform well without any positional encodings on large graphs, we reintroduce structural encodings here because UGDA demands sensitivity to structural shifts between domains. We also depart from SGFormer in our formulation to handle domains with different dimensions ($N^t \neq N^s$), and *sequentially* cascading the multi-layer Transformer and GCN (instead of linearly combining them in parallel Wu et al. (2023b)).

## 3.3 Cross-Attention for Source-Target Alignment

To bridge the domain gap in the TUGDA architecture, we introduce a cross-attention module that explicitly aligns source and target node representations. The intuition is to allow information flow between the two graphs so that target nodes can directly attend to relevant labeled source nodes and learn domain-invariant features. We implement cross-domain alignment as an additional transformer-style attention layer operating *across* the output of source transformer, and the target domain. We compute the output of cross-attention output $Z_{\text{XTrans}}^t$ with $Q^t = f_Q(z^{t(0)})$, $K^s = f_K(Z_{\text{Trans}}^s)$, and $V^s = f_V(Z_{\text{Trans}}^s)$, where $Z_{\text{Trans}}^s$ represents the source transformer output. Unlike this cross-attention mechanism, the source encoder uses only self-attention layers with uniform input $h^{s(0)}$ across all projections.

This mechanism effectively learns topology-independent semantic alignment between source and target node distributions. Unlike A2GNN's asymmetric propagation where alignment is constrained by adjacency matrices, our cross-attention computes affinities $\alpha_{ij}$ across all source-target pairs regardless of edge connectivity, enabling semantic correspondences between nodes that may be structurally distant or unreachable through k-hop message passing. Like the source encoder, $Z_{\text{XTrans}}^t$ is then concatenated with the original self-attended features and passed to the GCN layer for neighborhood aggregation. By exchanging representations across domains before topological structure is incorporated, cross-attention discovers latent semantic factors independent of graph topology.

The attention matrix acts as a learnable transport plan mapping target nodes to semantically similar source nodes in feature space, minimizing inter-domain distances without requiring topological proximity. In practice, this module can be applied before the GCN layer and requires no domain-adversarial losses, achieving alignment through integrated architectural bias that decouples semantic correspondence from structural constraints.

## 3.4 SEQUENTIAL INTEGRATION OF GCN AND TRANSFORMER

The final stage of our model is a Graph Convolutional Network (GCN) layer that operates on each graph's adjacency structure to inject local neighborhood information and further smooth the learned representations. Unlike previous approaches Wu et al. (2023b), dealing with the trade-off between the GCN and transformer, we feed the transformer output, after self-attention and cross-attention of source and target respectively, directly into a GCN.

The sequential flow is: global context $\rightarrow$ local propagation for the source encoder, and global context $\rightarrow$ aligned features $\rightarrow$ local propagation for the target encoder. Concretely, for each domain of source (s) and target (t), $d = \{s, t\}$, and each node $i \in V^d$, we form an augmented feature vector $\tilde{x}_i^d = [x_i^d \| Z_{\text{Trans},i}^d]$, concatenating the transformer's output for node $i$ with its original input features $x_i^d$. This preserves the raw feature signal and low-level information that might have been diminished during attention, providing the subsequent GCN more to work with. A single GCN layer is then applied:

$$Z_{\text{GCN}}^d = \sigma \left( \tilde{A}^d \cdot \tilde{X}^d \cdot W_{\text{GCN}} \right), \quad d = s, t, \tag{2}$$

where $\tilde{A}^d$ is the normalized adjacency matrix and $\tilde{X}^d$ is the augmented feature matrix of domain d, and $W_{\text{GCN}}$ is the learnable GCN weight matrix shared between source and target encoders. Equation 2, implements a standard first-order GCN convolution with asymmetric propagations between the source and target encoders Liu et al. (2024a).

By applying the GCN after the global attention, we let the model refine the globally aligned features with domain-specific local context, which is crucial for classification on each graph. The output of the source encoder, $Z_{GCN}^s$, is then fed to a classifier (i.e. a GCN layer followed by softmax) to predict source node labels.

## 3.5 OPTIMIZATION OBJECTIVE

To transfer knowledge from a labeled source to an unlabeled target, we train the model using the source graph's labels with a standard cross-entropy loss term $\mathcal{L}_{\text{cls}}$ between true and predicted label ($Y^s$ and $\hat{Y}^s$). To prevent overfitting to source-specific features and encourage domain-invariance, we incorporate an alignment regularizer: we minimize the discrepancy between source and target feature distributions in the shared latent space. In practice, one can use an *MMD (maximum mean discrepancy)* loss, $\mathcal{L}_{\text{align}}$, or similar on the GCN outputs to penalize differences Liu et al. (2024a):

$$\mathcal{L} = \mathcal{L}_{\text{cls}}(Y^s, \hat{Y}^s) + \alpha \, \mathcal{L}_{\text{align}}(Z_{\text{GCN}}^s, Z_{\text{GCN}}^t), \tag{3}$$

with a trade-off parameter $\alpha$. $Z_{GCN}^d$, $d = s, t$ is defined in equation 2. Specifically, alignment loss encourages closer source and target mean embeddings, further aligning the domains, $\mathcal{L}_{\text{align}} = \left| \frac{1}{N^s} \sum_{i \in V^s} \phi(z_{GCN,i}^s) - \frac{1}{N^t} \sum_{j \in V^t} \phi(z_{GCN,j}^t) \right|^2$. Here, $\phi(\cdot)$ is an implicit feature map of a reproducing kernel (e.g., RBF) and $N^s$, $N^t$ are source and target graph sizes.

In summary, our sequential Transformer-GCN architecture decouples semantic alignment from topological refinement: transformers first establish topology-independent cross-domain correspondences via self-attention and cross-attention, then GCN layers refine these semantically-aligned features with local structural context. This contrasts with structure- mediated methods like A2GNN where topology constrains alignment throughout. Our theoretical analysis (Section 4) proves this sequential design yields tighter domain adaptation bounds. The time comparison of our method against other baselines is available in the Appendix.

# 4 THEORETICAL ANALYSIS

In this section, we analyze the proposed Transformer-GCN framework through domain adaptation, providing formal insight into why our sequential architecture with cross-attention and smoothing should yield a low target error. Our theoretical analysis establishes architectural guarantees rather than optimization convergence results. We ground our discussion in the classic unsupervised domain adaptation setting and assume the standard **graph covariate shift** condition:

**Assumption 1** (Covariate Shift). $P_S(G) \neq P_T(G)$ *while* $P_S(Y \mid G) = P_T(Y \mid G)$.

This assumption guarantees the existence of a hypothesis that can perform well on both domains. Our contribution is demonstrating what can be achieved with the right architecture, validated empirically. Our Wasserstein-based analysis differs from recent UGDA theory in alignment domain and coupling mechanism. SpecReg (You et al. (2023)) aligns spectral filters via eigendecomposition, degrading when eigenstructures differ. HGDA (Fang et al. (2025b)) uses PAC-Bayes bounds with Wasserstein decomposition to prescribe alignment of specific filtered signals through topology-constrained graph operators. Our framework operates in learned feature space: Lemma 1 shows cross-attention represents soft transport couplings, and Theorem 2 proves sequential Transformer→GCN achieves tighter bounds. Unlike spectral or prescribed-filter methods, our attention learns semantic correspondence across all node pairs independent of graph structure, which GCN then refines structurally. The detailed discussion on the comparison between these methods is available in the Appendix.

### 4.0.1 DOMAIN ADAPTATION RISK DECOMPOSITION

Let $h \in \mathcal{H}$ denote a hypothesis mapping graphs to labels. The probability that a hypothesis $h$ disagrees with a predictive labeling function like $f_T$ on the target graph is the target risk, denoted by $\epsilon_T(h) := \epsilon_T(h, f_T)$. We seek to bound the target risk $\epsilon_T(h)$ of a hypothesis (classifier) in terms of the source risk and a measure of domain divergence:

**Theorem 1** (Domain Adaptation Bound Ben-David et al. (2010)). *Under standard assumptions, for any $h \in \mathcal{H}$,*

$$\epsilon_T(h) \leq \epsilon_S(h) + \Delta\big(P_S(Z), P_T(Z)\big) + \lambda^*, \tag{4}$$

*where $\epsilon_S(h)$ is the empirical source risk, $\Delta$ is a domain divergence measure (e.g., Wasserstein-1) between the distributions of source and target representations, $P_S(Z)$ and $P_T(Z)$, respectively, and $\lambda^*$ is the error of the ideal joint hypothesis.*

Intuitively, inequality 4 shows that the target error is bounded by the source error plus a penalty for how "far apart" the domains are in the representation space on which $\mathcal{H}$ operates. In this paper, we minimize the divergence term by aligning feature distributions and minimize $\epsilon_S(h)$ by learning source domain, all while keeping $\lambda^*$ small (i.e., assuming the same hypothesis is being applied on source and target tasks).

Assuming $h = c \circ f$ with a $C$-Lipschitz feature extractor $f$ and classifier $c$, and using Wasserstein-1 distance Villani et al. (2008), inspired by Theorem 1 You et al. (2023) we conjecture that the bound (4) becomes:

$$\epsilon_T(h) \leq \epsilon_S(h) + 2CW_1(P_S(Z), P_T(Z)) + \lambda^*. \tag{5}$$

### 4.0.2 ARCHITECTURE WITH TRANSFORMER AND GCN

We define a model consisting of a Transformer followed by a GCN, where $f_{\text{Trans}}^s(X^s) = Z^s$ is the source Transformer, and $f_{\text{Trans}}^t(X^t, Z^s) = Z^t$ is the target Transformer using cross-attention from $Z^s$.

**Cross-Attention Reduces Divergence Between Domains:** The practical implication of equation 5 is that when the Wasserstein distance between learned feature representations of source and target is smaller, the target error is closer to the source error. Our model promotes exactly this closeness: the cross-attention mechanism effectively learns to reduce the distance between $P_S(Z)$ and $P_T(Z)$ by aligning target features with source feature representations.

**Lemma 1** (Domain Alignment through Cross-Attention). *Let $\alpha_{ij}$ be the cross-attention weight from target node $i$ to source node $j$. $Z_i^t$ and $Z_j^t$ are transformer representations of node $i$ in Target and*

*node $j$ in Source, respectively. Then*

$$W_1(P_S(Z^s), P_T(Z^t)) \leq \mathbb{E}_{Z^t \sim P_T} \left[ \sum_j \alpha_{ij} \|Z^t - Z_j^s\| \right] + \epsilon_{coupling}, \quad (6)$$

*where $\|.\|$ represents L1 norm and $\epsilon_{coupling} = \max_j \left| \frac{1}{N^t} \sum_i \alpha_{ij} - \frac{1}{N^s} \right|$ is the source and target coupling violation. When $Z_i^t = Z_j^s$ for some j with $\alpha_{ij} = 1$, this distance is minimized.*

In an ideal scenario where for every target node $i$, the attention $\alpha_{ij}$ places weight on a source neighbor $j$ of the same class, feature distributions would align perfectly (resulting in $W_1 = 0$ in the latent space). While perfect alignment isn't achievable without labels, cross-attention provides a learnable transport that can significantly reduce distribution discrepancy in feature space, especially when combined with a loss like MMD that explicitly encourages overlap. We expect the second term in equation 5 to be small for our learned $f$ and the third term is typically small with the attentions distributed reasonably across source nodes, limiting the penalty on $\epsilon_T(h)$. This alignment is learned jointly with label prediction on source, so features are optimized to be both discriminative (low $\epsilon_S(h)$) and shared across domains (low divergence) – a balance that is suggested by classic DA theory to be necessary for low $\epsilon_T(h)$. You et al. (2023) Proof of Lemma 1 is available in Appendix.

**GCN Promotes Smoothness:** The GCN layer acts as a low-pass filter, enforcing local smoothness. We hypothesize that it reduces the model's sensitivity to high-frequency (noisy or domain-specific) components and enforces a Lipschitz constraint on $h$. Its Lipschitz constant $C_f$ can be bounded by the spectral norms of the propagation operator. Smoothness helps to control the divergence term in the adaptation bound.

**Theorem 2** (Transferability of Transformer+GCN). *Let $h^s = f_{GCN} \circ f_{Trans}^s$ and $h^t = f_{GCN} \circ f_{Trans}^t$. Suppose $f_{GCN}$ is $C_f$-Lipschitz and the Transformer achieves alignment such that $W_1(P_S(Z^s), P_T(Z^t)) = \delta$. Then*

$$\epsilon_T(h^t) \leq \epsilon_S(h^s) + 2C_f(\delta + \epsilon_{coupling}) + \mathcal{O}\left(\sqrt{\frac{1}{N_S}}\right). \quad (7)$$

*This bound is tighter than that of standard GCNs due to improved alignment (lower $\delta$) and regularized propagation (smaller $C_f$).*

Due to the cross-domain attention alignment with reasonable distribution, $\delta + \epsilon_{coupling}$ is small; thanks to the GCN smoothing, $C_f$ is also small; and we train to minimize $\epsilon_S(h^s)$ on the source. Therefore, all terms on the right side of inequality 7 are controlled, leading to a low target risk. In qualitative terms, our model learns a function that (1) fits the source data well, (2) doesn't change too abruptly with input perturbations or structural variations (Lipschitz/smooth), and (3) yields very similar feature distributions for source and target (aligned latent spaces). According to DA theory You et al. (2023); Chen et al. (2025), such a function would be transferred effectively. Proof of Theorem 2 is available in Appendix.

## 5 EXPERIMENTS

### 5.1 DATASETS AND EXPERIMENTAL SETUP

We evaluate our method, TUGDA, on three real-world citation graph datasets commonly used for cross-domain node classification: ACMv9 (A), Citationv1 (C), and DBLPv7 (D). Statistics of these datasets are summarized in Appendix. Following established practice in the literature, we conduct experiments across six cross-domain adaptation tasks: D → A, A → D, A → C, C → A, C → D, and D → C. We maintained consistent hyperparameter optimization across all methods. Results are reported as averages over five independent runs, using macro-F1 (Ma-F1) and micro-F1 (Mi-F1) scores.

### 5.2 BASELINES

To demonstrate the superiority of our method, we compare it against several SOTA methods in multiple categories: **Foundational Graph Representation Methods**: UDAGCN Wu et al. (2020a),

GRADE Wu et al. (2023a), StruRW Liu et al. (2023b), PairAlign Liu et al. (2024b); **Early Graph Domain Adaptation Approaches**: CDNE Shen et al. (2020b), AdaGCN Dai et al. (2022), SpecReg You et al. (2023), ACDNE Shen et al. (2020c); **Advanced Graph-Specific Domain Adaptation Models**: A2GNN Liu et al. (2024a), TDSS Chen et al. (2025), DGSDA Yang et al. (2025), DAUGNN Zhang & Fink (2025).

Table 1: Node classification performance (%) across six cross-domain adaptation tasks. Best results are bolded.

| Method | D→A | | A→D | | A→C | | C→A | | C→D | | D→C | | Avg | |
|---|---|---|---|---|---|---|---|---|---|---|---|---|---|---|
| | Ma-F1 | Mi-F1 | Ma-F1 | Mi-F1 | Ma-F1 | Mi-F1 | Ma-F1 | Mi-F1 | Ma-F1 | Mi-F1 | Ma-F1 | Mi-F1 | Ma-F1 | Mi-F1 |
| **Foundational Graph Representation Methods** | | | | | | | | | | | | | | |
| UDAGCN | 55.89 | 58.16 | 64.83 | 66.95 | 60.33 | 72.15 | 55.89 | 58.16 | 69.46 | 71.77 | 61.12 | 73.28 | 61.25 | 66.75 |
| GRADE | 59.35 | 63.72 | 63.03 | 68.22 | 72.52 | 76.04 | 59.35 | 63.72 | 70.02 | 73.95 | 69.32 | 74.32 | 65.60 | 70.00 |
| StruRW | 53.82 | 63.27 | 62.51 | 69.10 | 72.07 | 77.35 | 59.77 | 67.81 | 66.89 | 73.81 | 62.94 | 72.41 | 63.00 | 70.63 |
| PairAlign | 58.77 | 59.34 | 62.35 | 65.91 | 67.88 | 70.88 | 65.09 | 65.85 | 67.56 | 71.04 | 64.61 | 67.07 | 64.38 | 66.68 |
| **Early Graph Domain Adaptation Approaches** | | | | | | | | | | | | | | |
| CDNE | 70.45 | 69.62 | 69.24 | 71.58 | 76.83 | 78.76 | 70.45 | 69.62 | 71.34 | 74.36 | 77.36 | 78.88 | 72.61 | 73.80 |
| AdaGCN | 69.47 | 69.67 | 71.39 | 75.04 | 76.51 | 79.32 | 69.47 | 69.67 | 72.34 | 75.59 | 74.22 | 78.20 | 72.23 | 74.58 |
| SpecReg | 72.34 | 71.01 | 73.98 | 75.93 | 78.83 | 80.55 | 72.34 | 71.01 | 73.64 | 75.74 | 77.78 | 79.04 | 74.82 | 75.55 |
| ACDNE | 72.64 | 71.29 | 73.59 | 76.24 | 80.09 | 81.75 | 72.64 | 71.29 | 75.74 | 77.21 | 78.83 | 80.14 | 75.59 | 76.32 |
| **Advanced Graph-Specific Domain Adaptation Models** | | | | | | | | | | | | | | |
| A2GNN | 74.69 | 73.62 | 74.80 | 77.33 | 80.86 | 82.34 | 76.31 | 74.93 | 74.70 | 78.23 | 77.64 | 80.42 | 76.50 | 77.65 |
| TDSS | 75.37 | 73.88 | 75.39 | 78.36 | 81.03 | 82.66 | 76.04 | 74.72 | 74.76 | 78.11 | 78.71 | 80.93 | 76.88 | 78.11 |
| DGSDA | 74.67 | 72.87 | 73.91 | 76.33 | 81.28 | 82.65 | 76.47 | 75.08 | 75.73 | 77.67 | **80.72** | 82.09 | 77.13 | 77.78 |
| DAUGNN | 73.28 | 73.30 | 75.71 | 77.80 | 81.06 | 82.82 | 75.11 | 74.48 | 75.91 | 77.77 | 78.29 | 80.53 | 76.56 | 77.78 |
| **TUGDA (Ours)** | **76.26** | **75.10** | **76.65** | **78.41** | **81.84** | **83.17** | **76.85** | **75.57** | **76.63** | **78.3** | 80.27 | **82.21** | **78.08** | **78.80** |

## 5.3 Implementation Details

Following previous baseline settings Wu et al. (2020a); Shen et al. (2020a), we use 80% of the labeled source domain nodes for training, 20% for validation, with target domain nodes used for testing. All experiments are conducted on NVIDIA A100 GPUs, using PyTorchPaszke et al. (2017) and PyTorch Geometric libraryFey & Lenssen (2019).

We fine-tuned key hyperparameters within specific ranges: hidden dimensions for both GCN and transformer within $\{32, 64, 128, 256, 512\}$, learning rate within $\{0.01, 0.001, 0.005\}$, and propagation layer count for both source and target GCNS within $\{0, 5, 10, 20, 30, 40, 50\}$. The sensitivity of our model to the change of these hyperparameters is illustrated in Figure 2. Additionally, we assign $\beta = 0.5$ in equation 1 for all experiments. Large Language Models were used to improve the clarity and grammar of the manuscript.

## 5.4 Results and Analysis

Table 1 presents a comprehensive comparison of performances averaged over five trials for all evaluated methods across six standard cross-domain adaptation tasks for node classification. Highest performances are highlighted in bold and the second best ones are underlined. Standard deviation of results are available in Appendix. It is notable here that **Advanced Graph-Specific Domain Adaptation Models** deliver remarkable gains of up to **20%** on multiple tasks compared to **Foundational Graph Representation Methods**, suggesting insufficiency of simplistic pretext tasks or shallow encoders. Similarly, the traditional **Early Graph Domain Adaptation Approaches**, based on alignment with lack of structural awareness, are outperformed. This gap suggests that leveraging cross-domain relational information in a unified, end-to-end deep architecture is critical.

Our method, TUGDA, consistently outperforms SOTA domain adaptation baselines in 11 out of 12 evaluations, demonstrating its superior effectiveness. Notably, it surpasses strong recent baselines—particularly A2GNN—by a clear margin, highlighting its ability to capture both local and global structural dependencies. TUGDA excels in challenging transfer scenarios, such as adapting from denser to sparser citation domains (e.g., D → A), outperforming the next-best method (TDSS) by **+0.89%**, and shows symmetric strength in the reverse direction. It also achieves up to **+0.52%** improvement over DGSDA in the A → C task, underlining its effectiveness in aligning semantic

and structural patterns across heterophilic graphs. While DGSDA slightly outperforms TUGDA in Macro-F1 for the D → C task, TUGDA achieves the best Micro-F1, indicating more reliable predictions overall. TUGDA shows robustness across domain shifts. Its strong Micro-F1 and Macro-F1 indicate well-calibrated representations without bias toward dominant classes, unlike baselines that drop sharply in Macro-F1 on tasks like A → D and C → D—a key concern in imbalanced citation networks.

### 5.4.1 TUGDA Outperforms A2GNN on Generative Graphs

To address security and privacy constraints that preclude access to the original real-world source domain data, we introduce a novel evaluation framework using generative model-derived synthetic source graphs that preserve the statistical properties and feature distributions of the original data. We utilize the pretrained generative model of GraphMaker Li et al. (2023) and fine-tune it to generate the synthetic source domains (SynA, SynC, and SynD) with the most informative subset of original A, C, and D source features, respectively. Significant outperformance of our method, TUGDA, over A2GNN in Table 2 provides valuable insights into our method's superiority under controlled conditions and suggests the generalizability of our method to privacy-preserving synthetic data generation scenarios. This can serve as a viable proxy for evaluating domain adaptation techniques when access to sensitive real-world source domains is restricted.

Table 2: Performances (%) of A2GNN and TUGDA across synthetic sources to real-world target adaptations.

| Method | SynD → A | | SynA → C | | SynC → D | | SynD → C | | SynA → D | | SynC → A | | Avg | |
|---|---|---|---|---|---|---|---|---|---|---|---|---|---|---|
| | Macro | Micro | Macro | Micro | Macro | Micro | Macro | Micro | Macro | Micro | Macro | Micro | Macro | Micro |
| A2GNN | 57.86 | 61.84 | 58.52 | 62.92 | 56.33 | 59.77 | 62.22 | 63.28 | 64.87 | 66.41 | 54.73 | 54.70 | 59.09 | 61.49 |
| TDSS | 53.07 | 53.69 | 57.95 | 64.11 | 59.56 | 62.92 | 65.71 | 67.38 | 53.48 | 55.89 | 54.27 | 56.71 | 57.34 | 60.12 |
| DGSDA | **68.75** | **68.47** | **75.32** | **77.15** | 66.49 | 68.9 | 63.52 | 66.82 | 62.49 | 67.56 | 52.48 | 52.92 | 64.84 | 66.97 |
| TUGDA | 62.37 | 63.12 | 73.11 | 74.74 | **69.63** | **72.10** | 66.76 | 68.58 | 66.44 | 68.76 | 60.98 | 62.50 | **66.55** | **68.3** |

Table 2 reveals a critical finding: while real-to-real gains are modest due to benchmark saturation, synthetic-to-real adaptation shows substantially larger performance differentiation. TUGDA's average performance (66.55% Macro-F1, 68.3% Micro-F1) surpasses A2GNN by 7.46%/6.81% and achieves competitive results against DGSDA (64.84%, 66.97%), with notable advantages on SynC→D (+3.14%) and SynD→C (+3.24%). This validates our hypothesis: cross-attention's topology-independent alignment becomes critical when source-target structural correspondence is weak—precisely when synthetic graphs approximate statistical properties without preserving exact topology.

### 5.4.2 Comparison with Concurrent Work

We compare TUGDA with GAA (Fang et al., 2025a), a concurrent work that addresses UGDA through attribute-driven dual-channel GNNs. Since GAA has not released code and reports only accuracy metrics, we compare on the six overlapping citation network transfers where both methods provide results. For our multiclass single-label task, accuracy and micro-F1 are mathematically equivalent, enabling direct comparison.

Table 3: Comparison with GAA on Citation Networks.

| Method | D→A | A→D | A→C | C→A | C→D | D→C | Avg |
|---|---|---|---|---|---|---|---|
| GAA | **75.4** | **78.9** | 82.4 | **78.2** | 77.1 | 79.8 | 78.63 |
| TUGDA | 75.1 | 78.41 | **83.17** | 75.57 | **78.3** | **82.21** | **78.79** |

TUGDA achieves superior average performance (78.79% vs. 78.63%) and wins 3/6 transfers, with notable improvements on D→C (+2.41%) and C→D (+1.20%). While GAA uses attribute-driven dual-channel GNNs, TUGDA introduces novel cross-attention for domain alignment with theoretical guarantees (Theorem 2) and demonstrates robust generalization to synthetic sources (Table 2).

## 5.5 ABLATION STUDY

We perform an extensive ablation study to investigate the contributions of key components of our framework and evaluate it in different scenarios.

**Modules Effectiveness Study**

Figure 2 (a) demonstrates the impact of removing individual components from our transformer framework. We compare our full method against versions without centrality positional encoding (w/o CPE), without spatial positional encoding (w/o SPE), without all positional encodings (w/o PE), and without cross-attention across domains (w/o XAttn). The Macro-F1 scores are normalized over A2GNN.

Beyond the superiority of our method, the pronounced drops in C→D and A→D when positional encodings, especially CPE, are removed highlight their critical role. This underscores that DBLPv7 performance is particularly sensitive to structural cues in attention. The consistent gains with all components included confirm the necessity of both positional encodings and cross-attention.

**Parameter Sensitivity**

We analyze the sensitivity of our model to key hyperparameters, specifically hidden dimensions of GCN and transformer, and propagation layer count for both source and target domain in Figures 2 (b) and (c) in the A → C scenario.

The best performance occurs with 128 GCN hidden dimensions (Figure 2 (b)). Additionally, consistent with A2GNN, Figure 2 (c) demonstrates that the best performance happens with the smallest number of propagation layers for the source domain. However, the optimal number of propagation layers for the target domain is scenario-specific (Illustrated in the Appendix).

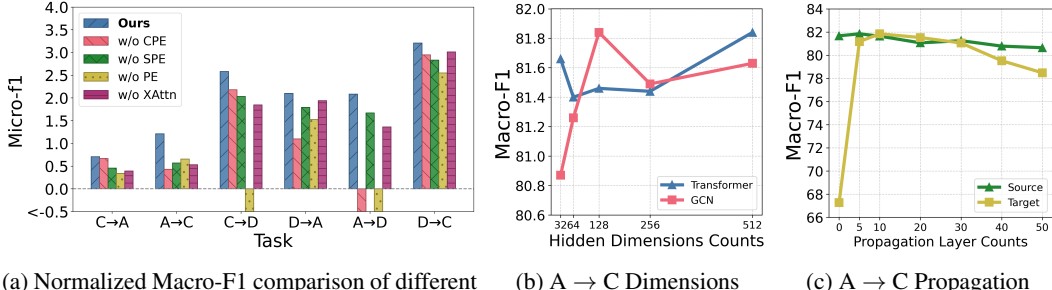

(a) Normalized Macro-F1 comparison of different versions of our method.

(b) A → C Dimensions

(c) A → C Propagation

Figure 2: Component contributions and the whole model parameter sensitivity analysis

## 6 CONCLUSION

In this work, we introduce a novel transformer-based framework for UGDA that unifies global structural modeling through transformers with local topology refinement utilizing asymmetric GCNs in both homophilic source and target domains. Our theoretical analysis investigates that the proposed cross-attention mechanism explicitly reduces source and target domain divergence. Our extensive experiments showcase consistent and substantial performance gains over SOTA baselines across both standard benchmarks and synthetic graphs generated by a foundational model. Critically, our framework's robust performance on synthetic source domains addresses practical deployment constraints where data privacy limits access to real source graphs, enabling TUGDA application in sensitive domains like healthcare, finance, and corporate networks, establishing it as a generalizable solution for graph domain adaptation challenges. While foundation model-based domain generalization represents an important 'train-once, generalize-everywhere' direction, our target-specific DA framework provides complementary strengths through higher accuracy via target distribution adaptation and formal performance guarantees (Theorem 2), suggesting hybrid approaches as promising future work. Going ahead, we will extend our domain adaptation studies to non-citation graphs, including biological networks and vision graphs. Further, we intend to develop analytical bounds on target error when distribution shifts between domains are covariate and semantic shifts.

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

# 7 APPENDIX

## 7.1 ALGORITHM DESCRIPTION

In this section, the algorithm of the proposed method, TUGDA, is presented in Algorithm 1.

---

**Algorithm 1** TUGDA: Transformer-based Unsupervised Graph Domain Adaptation

---

**Require:** Source graph $G^s = (V^s, E^s, X^s, Y^s)$, Target graph $G^t = (V^t, E^t, X^t)$
**Ensure:** Target predictions $\hat{Y}^t$
 1: **Initialize:** Transformer parameters $f_Q, f_K, f_V$; GCN parameters $W_{GCN}$; Classifier $W_{cls}$
 2: **for** each training epoch **do**
 3:     // **Positional Encoding**
 4:     **for** domain $d \in \{s, t\}$ **do**
 5:         Apply centrality encoding using Eq. (1)
 6:         Compute spatial positional encodings for Eq. (2)
 7:     **end for**
 8:     // **Self-Attention for Both Domains**
 9:     **for** domain $d \in \{s, t\}$ **do**
10:         Compute $Q^d, K^d, V^d$ using Eq. (3)
11:         Apply transformer attention using Eq. (4) $\rightarrow Z^d_{Trans}$
12:     **end for**
13:     // **Cross-Attention (Target $\leftarrow$ Source)**
14:     $Q^t = f_Q(Z^t_{Trans}), K^s, V^s = f_K(Z^s_{Trans}), f_V(Z^s_{Trans})$
15:     Apply cross-attention mechanism $\rightarrow Z^t_{XTrans}$
16:     // **Sequential GCN Integration**
17:     Concatenate features: $\tilde{X}^s = [X^s \| Z^s_{Trans}], \tilde{X}^t = [X^t \| Z^t_{XTrans}]$
18:     Apply GCN using Eq. (5) with shared weights $\rightarrow Z^s_{GCN}, Z^t_{GCN}$
19:     // **Loss Computation and Optimization**
20:     $\hat{Y}^s = \text{softmax}(Z^s_{GCN} \cdot W_{cls})$
21:     Compute classification and alignment losses using Eq. (6)
22:     Update parameters: $\theta \leftarrow \theta - \eta \nabla_\theta \mathcal{L}_{total}$
23: **end for**
24: // **Inference**
25: **return** $\hat{Y}^t = \text{softmax}(Z^t_{GCN} \cdot W_{cls})$

---

## 7.2 ADDITIONAL EXPERIMENTAL DETAILS

Table 4 presents the statistics of the real-world datasets used in our framework evaluation. The datasets are available at https://github.com/Meihan-Liu/24AAAI-A2GNN.

Table 4: Statistics of the three real-world citation graphs.

| Graph | #Nodes | #Edges | #Attr. | #Labels | Density |
|---|---|---|---|---|---|
| ACMv9 (A) | 9,360 | 15,556 | 6,775 | 5 | 0.00036 |
| Citationv1 (C) | 8,935 | 15,098 | 6,775 | 5 | 0.00038 |
| DBLPv7 (D) | 5,484 | 8,117 | 6,775 | 5 | 0.00054 |

The results of evaluation experiments on the above datasets compared to other DA baselines are presented in Table 5 with the standard deviations included.

Figure 3 compares our framework in different scenarios, excluding different components of the method.

We analyze the sensitivity of our model to key hyperparameters, specifically hidden dimensions of GCN and transformer, and propagation layer count for both source and target domain in Figure 4 for D $\rightarrow$ A and A $\rightarrow$ C scenarios.

Table 5: Node classification performance (%) across six cross-domain adaptation tasks. Best results are bolded.

| Method | D→A | | A→D | | A→C | | C→A | | C→D | | D→C | |
|---|---|---|---|---|---|---|---|---|---|---|---|---|
| | Ma-F1 | Mi-F1 | Ma-F1 | Mi-F1 | Ma-F1 | Mi-F1 | Ma-F1 | Mi-F1 | Ma-F1 | Mi-F1 | Ma-F1 | Mi-F1 |
| **Domain Adaptation Methods** | | | | | | | | | | | | |
| A2GNN | 74.69 ± 0.3 | 73.62 ± 0.2 | 74.80 ± 0.5 | 77.33 ± 0.5 | 80.86 ± 0.2 | 82.34 ± 0.2 | 76.31 ±0.2 | 74.93 ± 0.3 | 74.70 ± 0.7 | 78.23±0.4 | 77.64±0.5 | 80.42 ±0.4 |
| TDSS | 75.37 ± 0.01 | 73.88 ± 0.06 | 75.39 ± 0.6 | 78.36 ± 0.2 | 81.03 ± 0.2 | 82.66 ± 0.2 | 76.04 ± 0.09 | 74.72 ± 0.09 | 74.76 ± 0.2 | 78.11 ± 0.4 | 78.71 ± 0.5 | 80.93 ± 0.1 |
| DGSDA | 74.67 ± 0.3 | 72.87 ± 0.2 | 73.91 ± 0.4 | 76.33 ± 0.4 | 81.28 ± 0.3 | 82.65 ± 0.3 | 76.47 ± 0.2 | 75.08 ± 0.2 | 75.73 ± 0.3 | 77.67 ± 0.4 | **80.72 ± 0.3** | 82.09 ± 0.3 |
| **TUGDA (Ours)** | **76.26 ± 0.2** | **75.10 ± 0.1** | **76.65 ± 0.3** | **78.41 ± 0.1** | **81.84 ± 0.2** | **83.17 ± 0.2** | **76.85 ± 0.3** | **75.57 ± 0.3** | **76.63 ± 0.6** | **78.3 ± 0.3** | 80.27 ± 0.4 | **82.21 ± 0.4** |

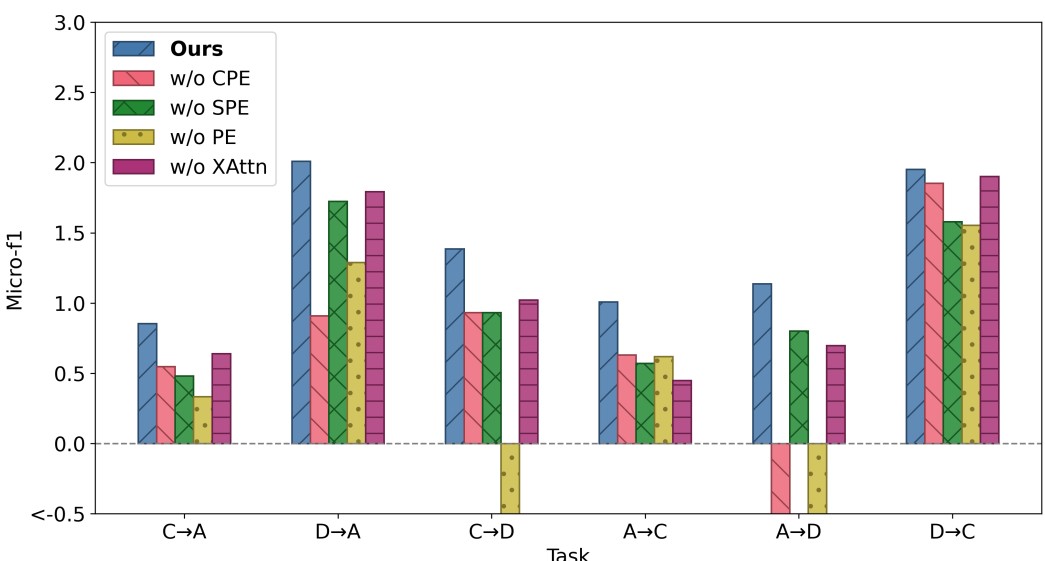

Figure 3: Effectiveness of different components of TUGDA (measured by Micro-F1).

In both scenarios, the best performances occur with 128 GCN hidden dimensions, but the optimal transformer hidden dimension varies by scenario. (Figure 4 (a) and (c)) Additionally, consistent with A2GNN, Figures 4 (b) and (d) demonstrate that the best performances happen with the smallest number of propagation layers for the source domain. However, the optimal number of propagation layers for the target domain is scenario-specific.

## 7.3 RELATED WORK

UGDA tackles distribution shifts across graph domains. General-purpose encoders such as UDAGCN, GRADE, StruRW, and PairAlign use self-supervised pretraining and structural matching. Transfer Learning methods like CDNE, AdaGCN, SpecReg, and ACDNE align domains through adversarial training, discrepancy minimization, or spectral regularization, but often overlook topology preservation. Recent UGDA models exploit direct structural adaptation: A2GNN aligns distributions via **asymmetric propagation layers** for source and target domains; TDSS reduces structural noise and stabilizes representation learning via Laplacian smoothing and neighborhood sampling; DGSDA incorporates Bernstein polynomial approximation for aligning graph spectral filters to avoid expensive eigen-decompositions; and DAUGNN adapts structure-aware message passing through iterative propagation and integrates alignment into the unfolding process.

While effective, these models remain limited by GNNs' local receptive fields. **Transformer-based graph models** capture global interactions: Graphormer encodes structural information into attention scores via shortest paths and centrality, while SGFormer provides scalable global attention without positional encodings. Our framework, **TUGDA**, combines SGFormer's efficient attention with Graphormer's structural encodings, then refines representations through an A2GNN-inspired asymmetric GCN. This hybrid design achieves joint global-local adaptation, surpassing prior UGDA baselines.

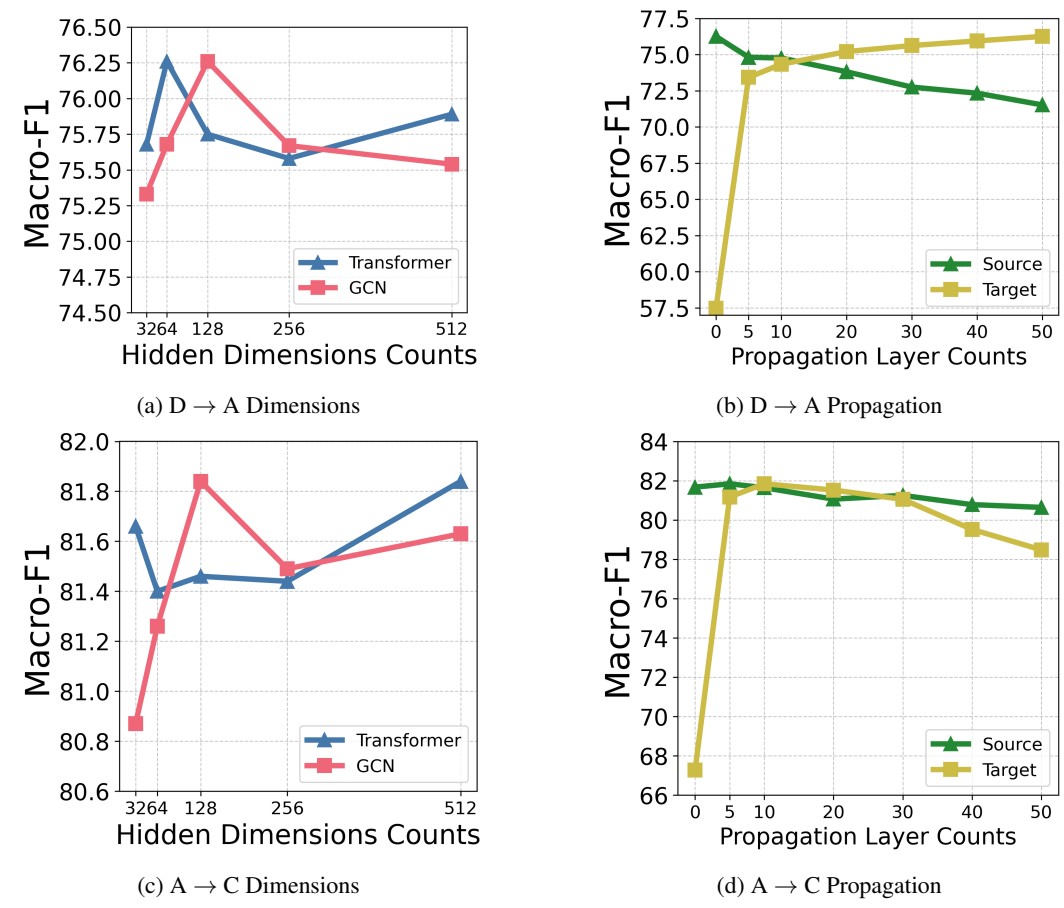

Figure 4: Dimensions and propagation layers' sensitivity.

## 7.4 THEORETICAL ANALYSIS PROOFS

In this section, the proofs of the theoretical analysis from the main paper are elaborated.

**Lemma 2** (Domain Alignment via Cross-Attention). *Let $\alpha_{ij}$ be the cross-attention weight from target node $i$ to source node $j$. Then:*

$$W_1(P_S(Z^s), P_T(Z^t)) \leq \delta_{align} + \epsilon_{coupling} \tag{8}$$

*where:*

$$\delta_{align} = \mathbb{E}_{i \sim P_T} \left[ \sum_j \alpha_{ij} \| Z_i^t - Z_j^s \| \right] \tag{9}$$

$$\epsilon_{coupling} = \max_j \left| \frac{1}{N^t} \sum_i \alpha_{ij} - \frac{1}{N^s} \right| \tag{10}$$

*Proof.* The Wasserstein-1 distance is defined as:

$$W_1(P_S, P_T) = \inf_{\gamma \in \Pi(P_S, P_T)} \mathbb{E}_{(x,y) \sim \gamma}[\|x - y\|] \tag{11}$$

where $\Pi(P_S, P_T)$ is the set of all joint distributions with marginals $P_S$ and $P_T$.

We define a pseudo-coupling $\hat{\gamma}$ using cross-attention weights:

$$\hat{\gamma}(Z_j^s, Z_i^t) = P_T(Z_i^t) \cdot \alpha_{ij} \tag{12}$$

We verify marginal constraints for this pseudo-coupling:

For the target marginal:

$$\sum_j \hat{\gamma}(Z_j^s, Z_i^t) = \sum_j P_T(Z_i^t)\alpha_{ij} = P_T(Z_i^t)\sum_j \alpha_{ij} \tag{13}$$
$$= P_T(Z_i^t)$$

(since softmax attention satisfies $\sum_j \alpha_{ij} = 1$)

For the source marginal, $P_S(Z_j^s)$:

$$\sum_i \hat{\gamma}(Z_j^s, Z_i^t) = \sum_i P_T(Z_i^t)\alpha_{ij} \tag{14}$$

If we assume uniform target distribution for simplicity: $P_T(Z_i^t) = 1/N^t$, then:

$$\sum_i \hat{\gamma}(Z_j^s, Z_i^t) = \frac{1}{N^t}\sum_i \alpha_{ij} \tag{15}$$

This should equal $P_S(Z_j^s) = 1/N^s$ if the attentions are normalized across source nodes, but in general, we have a coupling violation:

$$\left| \frac{1}{N^t}\sum_i \alpha_{ij} - \frac{1}{N^s} \right| \le \epsilon_{\text{coupling}} \tag{16}$$

We bound Wasserstein distance using the pseudo-coupling:

$$W_1(P_S, P_T) \le \mathbb{E}_{(Z^s, Z^t)\sim\hat{\gamma}}[\|Z^s - Z^t\|] + \text{coupling correction} \tag{17}$$
$$= \sum_{i,j} \hat{\gamma}(Z_j^s, Z_i^t)\|Z_j^s - Z_i^t\| + \epsilon_{\text{coupling}} \tag{18}$$
$$= \sum_{i,j} P_T(Z_i^t)\alpha_{ij}\|Z_i^t - Z_j^s\| + \epsilon_{\text{coupling}} \tag{19}$$
$$= \sum_i P_T(Z_i^t)\left(\sum_j \alpha_{ij}\|Z_i^t - Z_j^s\|\right) + \epsilon_{\text{coupling}} \tag{20}$$
$$= \mathbb{E}_{Z^t\sim P_T}\left[\sum_j \alpha_{ij}\|Z^t - Z_j^s\|\right] + \epsilon_{\text{coupling}} \tag{21}$$
$$= \delta_{\text{align}} + \epsilon_{\text{coupling}} \tag{22}$$

$\square$

**Theorem 3** (Transferability of Transformer+GCN). *Let* $h^s = f_{\text{GCN}} \circ f_{\text{Trans}}^s$ *and* $h^t = f_{\text{GCN}} \circ f_{\text{Trans}}^t$. *Suppose* $f_{\text{GCN}}$ *is* $C_f$-*Lipschitz and the Transformer achieves alignment such that* $W_1(P_S(Z^s), P_T(Z^t)) = \delta_{align} + \epsilon_{coupling}$. *Then:*

$$\epsilon_T(h^t) \le \epsilon_S(h^s) + 2C_f(\delta_{align} + \epsilon_{coupling}) + \mathcal{O}\left(\sqrt{\frac{1}{N_S}}\right) \tag{23}$$

*Proof.* Starting with the given domain adaptation bound using the Wasserstein distance:

$$\epsilon_T(h) \le \epsilon_S(h) + 2CW_1(P_S(Z), P_T(Z)) + \lambda^* \tag{24}$$

For our architecture, the representations fed to the final classifier are:

- Source: $Z^s = f_{\text{Trans}}^s(X^s)$

- Target: $Z^t = f_{\text{Trans}}^t(X^t, Z^s)$ (with cross-attention)

And the final hypotheses are:

- $h^s = f_{\text{GCN}}(Z^s, A^s)$

- $h^t = f_{\text{GCN}}(Z^t, A^t)$

By applying the GCN Lipschitz bound, The Wasserstein distance between final outputs is bounded by:

$$
\begin{aligned}
W_1(P_S(h^s), P_T(h^t)) &\leq C_f \cdot W_1(P_S(Z^s), P_T(Z^t)) \\
&= C_f(\delta_{\text{align}} + \epsilon_{\text{coupling}})
\end{aligned}
\tag{25}
$$

This uses the fact that Lipschitz functions can only increase Wasserstein distance by at most the Lipschitz constant, and applies our refined cross-attention bound.

Under the covariate shift assumption $P_S(Y|G) = P_T(Y|G)$, we have $\lambda^* = 0$ in the ideal case.

The finite sample term comes from empirical risk estimation:

$$
|\epsilon_S(h^s) - \hat{\epsilon}_S(h^s)| = \mathcal{O}\left(\sqrt{\frac{1}{N_S}}\right)
\tag{26}
$$

Combining all terms:

$$
\epsilon_T(h^t) \leq \epsilon_S(h^s) + 2C_f(\delta_{\text{align}} + \epsilon_{\text{coupling}}) + \mathcal{O}\left(\sqrt{\frac{1}{N_S}}\right)
\tag{27}
$$

$\square$

## 7.5 RUNTIME ANALYSIS

Although computational efficiency was not a primary focus of this work, we provide runtime measurements for transparency and to address scalability concerns. All experiments were conducted on NVIDIA A100 GPUs. We report wall-clock training time (in seconds) until convergence for representative transfer scenarios.

Table 6: Training time comparison (in seconds) on NVIDIA A100 GPU.

| Transfer | A2GNN | TUGDA | TDSS | DGSDA |
|----------|-------|-------|------|-------|
| A→C | 170 | 1150 | 910 | 296 |
| D→A | 255 | 1855 | 878 | 264 |

TUGDA incurs approximately 6-7× training overhead compared to A2GNN, primarily due to transformer self-attention and cross-domain attention computations. However, this overhead grows **linearly with target graph size** $N_t$, not quadratically, due to our factorized attention mechanism that avoids computing full $N_s \times N_t$ attention matrices.

The computational cost is justified by consistent accuracy improvements across all benchmarks (Tables 1, 2 in main paper). For applications where prediction quality is prioritized over training efficiency, TUGDA provides a favorable accuracy-efficiency trade-off. Future work could explore sparse attention mechanisms or knowledge distillation to improve computational efficiency for large-scale graphs.

## 7.6 THEORETICAL COMPARISON WITH RELATED WORK

SpecReg (You et al. (2023)) minimizes spectral filter divergence via explicit eigen decomposition of graph Laplacians, aligning graphs in spectral domain. This assumes spectral similarity—when eigen

structures differ drastically (D→A: 1.5× density), the bound degrades. TUGDA aligns in learned feature space Z via attention, decoupled from spectral properties. Our bound (Theorem 2) holds under spectral divergence since cross-attention learns semantic coupling from representations, not eigen decompositions.

HGDA (Fang et al. (2025b)) uses Wasserstein analysis to decompose and prescribe which graph operators to apply. We use it to prove our architecture can learn optimal semantic couplings through standard supervised training. When structural assumptions hold, HGDA's fixed filters may be more sample-efficient. When topology-structure correspondence breaks down, our learned couplings maintain effectiveness versus HGDA's applicability being limited to graphs with clear homophily decompositions.

