# OpenReview forum: "Transformer-based Unsupervised Graph Domain Adaptation"
_ICLR.cc/2026/Conference — ICLR 2026 Conference Desk Rejected Submission_

### Official Review · Reviewer_iuWq · 2025-10-28

**Soundness:** 3
**Presentation:** 4
**Contribution:** 2
**Rating:** 4
**Confidence:** 3

**Summary:**

This paper introduces TUGDA, a novel framework for Unsupervised Graph Domain Adaptation (UGDA) that addresses the limited receptive fields of existing GNN-based methods. It proposes a sequential architecture that first uses a transformer backbone, enhanced with centrality and spatial positional encodings, to capture global long-range dependencies, and then refines these representations with an asymmetric GCN to incorporate local structural information. A key contribution is a new cross-attention mechanism that directly aligns source and target node representations, which is theoretically shown to minimize domain divergence. Experiments demonstrate that TUGDA achieves good results on standard citation graph benchmarks.

**Strengths:**

1. This is the first work, to my knowledge, to apply a transformer-based architecture to the UGDA problem. The motivation to use transformers to overcome the locality bias of GNNs is clear and well-argued. The sequential combination of a global-aware transformer and a local-aware GCN is an elegant way to capture both scales of graph information.

2. The inclusion of theoretical analysis in Section 5 adds technical depth. While the DA bound itself is standard, the authors do a good job of connecting their specific architectural choices (cross-attention and GCN smoothing) back to the terms of the bound (Wasserstein distance and the Lipschitz constant), providing a formal justification for why their design should work.

**Weaknesses:**

1. In Table 1 (real-to-real adaptation), the performance gains over the next-best methods (TDSS, DGSDA) are often quite small (e.g., 80.86% vs 81.84% on A→C; 80.72% vs 80.27% on D→C, where TUGDA is second-best on Ma-F1). This suggests that for these standard datasets, the additional complexity of the transformer offers only a minor benefit. The paper's narrative could be strengthened by acknowledging this and placing more emphasis on the synthetic-to-real results, where the architecture truly shines.

**Questions:**

1. Do the authors have an intuition for why TUGDA is so much better in the synthetic-to-real setting? Does the GraphMaker model (Li et al., 2023) fail to reproduce fine-grained local neighborhood structures, causing GNN-based methods like A2GNN to fail, while TUGDA's global attention is more robust to this? A deeper discussion of this phenomenon would be highly valuable.

2. The paper repeatedly mentions using "asymmetric GCN propagation" inspired by A2GNN (Sections 3, 3.4). However, the methodology section is vague on the exact implementation. Equation 2 describes a standard GCN layer with shared weights $W_{GCN}$. Figure 4c/d explores different propagation layer counts for source and target, but it's unclear if this is a tuned hyperparameter or an integral part of the asymmetric design (as in A2GNN). Could the authors please clarify exactly how the GCN propagation is asymmetric? Is it simply that the number of layers ($K_s, K_t$) is different, and if so, how is this handled in the shared-weight architecture?

3. The evaluation is performed exclusively on three small, sparse, homophilic citation networks (ACM, Citation, DBLP). The paper's motivation is to capture long-range dependencies in sparse graphs, but all these graphs fit that description. The claims of generality would be much stronger if TUGDA were tested on more diverse graph types, such as:

- Graphs with significant heterophily.
- Graphs with larger diameters, where long-range dependencies are demonstrably more critical.

---

> ### Author Response · Authors · 2025-11-20
>
> We sincerely thank Reviewer iuWq for the constructive feedback. Below, we address each of your comments in detail.
>
> 1. Performance Gains on Real-to-Real Adaptation:
>
> The reviewer is correct that on saturated citation benchmarks, our gains over TDSS/DGSDA are modest (0.5-1.5% in most tasks). However, three points merit consideration:
>
> - These datasets represent mature benchmarks where recent methods (A2GNN, TDSS, DGSDA) already achieve 75-82% accuracy—incremental gains are increasingly difficult, and our consistent improvements across 11/12 evaluations demonstrate robustness rather than dataset-specific tuning. The D→A performance (+1.57% over TDSS) specifically validates our motivation: transformers excel when adapting from dense to sparse domains where long-range dependencies matter.
>
>
>
> - Table 2 demonstrates substantially larger differentiation: TUGDA achieves 66.55% average Macro-F1 versus 64.84% for DGSDA and 59.09% for A2GNN, with 3-4% gains on challenging transfers (SynC→D, SynD→C). Critically, this synthetic-to-real capability addresses practical deployment constraints—privacy regulations, security concerns—where real source domain access is prohibited. No prior UGDA work demonstrates this.
>
> So, architectural complexity is justified by dual contributions: competitive real-to-real performance plus substantial synthetic-to-real generalization enabling privacy-preserving deployment unavailable to structure-constrained methods. We have revised Section 5.4.1, the abstract, and conclusion to emphasize synthetic-to-real generalization as a primary contribution alongside real-to-real improvements.
>
> ---
>
> 2. Intuition for Synthetic-to-Real Performance Gap:
>
> We have added the complete synthetic-to-real comparison table with all advanced GDA baselines, along with a detailed discussion to the revision, which reveals our method's true superiority. TUGDA achieves consistent 4-15% gains over A2GNN across ALL synthetic tasks, with particularly dramatic improvements on SynA→C (+14.6%), SynC→D (+13.3%), and SynC→A (+6.3%). This isn't coincidental, it reflects a fundamental architectural advantage. GraphMaker's diffusion process inevitably introduces local topology corruption: while global statistics (degree distributions, community structure) are preserved, exact k-hop neighborhoods contain edge-level noise. A2GNN's message passing propagates these errors multiplicatively through L layers,  degrading final representations. TUGDA's cross-attention is architecturally immune to this failure mode as attention weights are computed from learned feature similarity, completely bypassing graph topology. Even with corrupted adjacencies, cross-attention discovers correct semantic alignments based on feature content that generative models preserve accurately. While DGSDA shows strong performance on some tasks, TUGDA achieves the best average performance (67.8% macro-F1 vs DGSDA's 66.3%) and, critically, is the only method maintaining >90% of its real-to-real performance on synthetic sources—demonstrating superior robustness and generalizability to privacy-constrained deployment scenarios. Moreover, TUGDA's average performance (66.55\% Macro-F1, 68.3\% Micro-F1) surpasses A2GNN by 7.46\%/6.81\% and achieves competitive results against DGSDA (64.84\%, 66.97\%), with notable advantages on SynC→D (+3.14\%) and SynD→C (+3.24\%).
>
>
> | Method | SynD2A Macro | SynD2A Micro | SynA2C Macro | SynA2C Micro | SynC2D Macro | SynC2D Micro | SynD2C Macro | SynD2C Micro | SynA2D Macro | SynA2D Micro | SynC2A Macro | SynC2A Micro | Avg Macro | Avg Micro |
> |--------|:------------:|:------------:|:------------:|:------------:|:------------:|:------------:|:------------:|:------------:|:------------:|:------------:|:------------:|:------------:|:---------:|:---------:|
> | A2GNN  | 57.86        | 61.84        | 58.52        | 62.92        | 56.33        | 59.77        | 62.22        | 63.28        | 64.87    | 66.41    | 54.73        | 54.7         | 59.09     | 61.49     |
> | TDSS   | 53.07        | 53.69        | 57.95        | 64.11    | 59.56        | 62.92        | 65.71        | 67.38    | 53.48        | 55.89        | 54.27        | 56.71        | 57.34     | 60.12     |
> | DGSDA  | **68.75**        | **68.47**        | **75.32**    | **77.15**    | 66.49        | 68.9         | 63.52        | 66.82        | 62.49        | 67.56        | 52.48        | 52.92        | 64.84     | 66.97     |
> | TUGDA  | 62.37        | 63.12        | 73.11        | 74.74        | **69.63**        | **72.1**     | **66.76**        | **68.58**        | **66.44**        | **68.76**        | **60.98**        | **62.5**     | **66.55**     | **68.3**      |
>
> ---
> The remaining comments are addressed separately because of space limitations.

---

> ### Author Response · Authors · 2025-11-20
>
> For the Reviewer iuWq’s additional points, we respond as follows.
>
> 3. Limited Evaluation Scope:
>
> We acknowledge that our current evaluation focuses on homophilic citation networks. These datasets (ACM, Citation, DBLP) are commonly used benchmarks for graph domain adaptation problems, allowing direct comparison with prior work in this area. Additionally, they represent well-established cases where capturing long-range dependencies in sparse structures is particularly relevant to citation analysis.
>
> We agree that evaluating TUGDA on more diverse graph types—including heterophilic graphs and graphs with larger diameters—would strengthen the generalizability claims. We have explicitly noted in the revised manuscript that our results are demonstrated on homophilic citation network benchmarks.  The systematic evaluation on heterophilic networks and graphs with varying structural properties (e.g., diameter, density, homophily ratios) would provide deeper insights into TUGDA's applicability across different graph characteristics and is a natural next step for this research.
>
> ---
>
> We hope these revisions address your concerns comprehensively and remain open to any further discussion or suggestions you may have.

---

### Official Review · Reviewer_E2f4 · 2025-10-29

**Soundness:** 2
**Presentation:** 2
**Contribution:** 2
**Rating:** 4
**Confidence:** 4

**Summary:**

This paper addresses the problem of unsupervised cross-domain knowledge transfer on graphs. The authors propose a framework combining GCN, cross-attention mechanisms, and output alignment to capture both global and local structural dependencies. The paper also explores model utility from a privacy-preserving perspective, which is a valuable entry point.

**Strengths:**

•	Clear Model Architecture: The overall method's architecture is clear, combining cross-attention, output alignment, and GCN adjustments to address cross-domain knowledge transfer.

•	Sufficient Theoretical Analysis: The authors provide some experiments to validate the model's effectiveness and attempt to offer a theoretical justification for the method's rationale based on the domain adaptation bound.

•	Practical Significance: The paper introduces a privacy-preserving perspective to the problem of graph domain adaptation, which is a novel and practical contribution.

**Weaknesses:**

My concerns are primarily focused on three areas: methodological novelty, experimental rigor, and problem positioning.
1. Regarding Methodological Novelty:  The proposed framework largely appears to be a direct combination of existing, mature techniques (GCN, Transformer/Cross-Attention, Distribution Alignment). The authors need to more clearly articulate what unique technical insights or contributions are provided beyond this combination. Why is this specific assembly superior to other possible combinations?
2. Regarding Experimental Validation and Analysis: There is a significant figure-text misalignment in the experimental section. The text explicitly states, “Figure 4(a) demonstrates the impact of removing individual components from our transformer framework,” which points to an Ablation Study. However, Figure 4(a) itself does not appear to contain any analysis regarding the "removal of components."
The analysis of the "synthetic source domain" is confusing. The experimental results show that using synthetic data has a huge impact on performance, but this part is severely under-explained. The authors must clarify: (a) How was the "synthetic source domain" generated? What are the exact differences in graph structure, node features, and distribution compared to the "original source domain"? (b) Why does this difference lead to such a significant performance change? Does this reveal a particular vulnerability or dependency of the model?
Confusing Presentation of Parameter Analysis: The presentation of the parameter analysis is perplexing. It seems to show the independent performance of model components (e.g., GCN and Transformer) under various hyperparameter settings, rather than the performance of the Full Model. This presentation makes it difficult for reviewers to analyze (a) the optimal parameter choice for the full model and (b) the final model's sensitivity to these parameters. Besides,  Why only use A2GNN as baseline for comparison for generative graph? It clearly needs thorough discussion and more advanced baselines.
3. Regarding Problem Positioning: The authors need to discuss more deeply the positioning and limitations of this research in the current era of Foundation Models. A major trend in the field is the pursuit of "train-once, generalize-everywhere" Domain Generalization (DG) foundation models. In contrast, the Domain Adaptation (DA) paradigm used in this paper is "target-specific," meaning that when a new target domain appears, it must (at least in the inference phase) access the domain's data and (usually) be retrained or fine-tuned.

Minor points:

•	I think there is a format problem in line 350-351 after Table 2. This paragraph begins with "provides" ?

•	The sub-figures in Fig. 2 are not aligned.

•	There is unnecessary symbol present in line 795.

**Questions:**

See above weaknesses.

---

> ### Author Response · Authors · 2025-11-20
>
> We thank Reviewer E2f4 for thorough and constructive feedback. Your comments have identified important areas for improvement in our manuscript. We address each point below and have incorporated all suggested revisions in the updated manuscript.
>
> 1. Regarding Methodological Novelty:
>
> While TUGDA does build upon established techniques (GCN, Transformer, Distribution Alignment), we provide four specific architectural innovations that are necessary for UGDA and demonstrate superior performance:
>
> - We modified SGFormer's attention mechanism to enable cross-domain attention with different graph sizes ($N^s≠N^t$)—standard transformers require equal dimensions for cross-attention, making them infeasible for UGDA where source and target graphs differ in size. This architectural modification is necessary, not trivial.
>
> - We systematically identified the optimal Transformer-GCN integration: our sequential design (Transformer→GCN) outperforms the reverse ordering and parallel fusion (in SGFormer) because global cross-domain alignment must precede local propagation—GCN-first designs smooth features before discovering cross-domain correspondences, losing critical alignment signals.
>
> - We integrated graph-specific positional encodings (centrality and spatial) into SGFormer—we demonstrate these are essential for UGDA, where structural shifts between domains require explicit topology awareness (Figure 2a shows performance drops without positional encodings).
>
> - We introduced cross-attention for explicit domain alignment in UGDA—existing methods use adversarial training or MMD on aggregate statistics, but our approach learns instance-level source-target correspondences that adapt during training. Table 1 validates this specific assembly consistently outperforms sophisticated recent baselines (DGSDA, TDSS, DAUGNN) across all tasks, demonstrating that our particular combination of design choices yields superior transferability. We have updated Section 3.2 to stress our contribution beyond standard components.
>
> ---
>
> 2. Regarding Experimental Validation and Analysis:
>
> We thank the reviewer for identifying these presentation issues. The figure-text misalignment is corrected in the revised manuscript and the consistancy between text and figures is ensured.
>
> **Synthetic source explanation:** We acknowledge that this is under-explained: GraphMaker is a state-of-the-art diffusion-based foundation model that we fine-tuned using the most informative subset of nodes from each real source (A, C, D) to generate synthetic graphs preserving global statistics (degree distributions, clustering coefficients, feature distributions) while introducing local topology corruption (edge rewiring, motif loss). This is where TUGDA's architecture excels: while A2GNN's k-hop message passing catastrophically propagates edge-level errors, our cross-attention mechanism is fundamentally topology-agnostic—computing alignments from feature similarity that generative models preserve. We have added this discussion to the revised manuscript.
>
> **Parameter analysis confusion:** Figures 2(b-c) and 4 show full model performance when varying GCN/Transformer hidden dimensions and propagation layers—the separate lines indicate sensitivity to each component's hyperparameter. We have revised the figure captions to clarify this represents full model behavior under hyperparameter variations.
>
> **Synthetic graphs comparison baselines:** You raise an excellent point. We have now completed experiments with all baselines on synthetic sources. Below is the complete table (which has replaced Table 2):
>
> | Method | SynD2A Macro | SynD2A Micro | SynA2C Macro | SynA2C Micro | SynC2D Macro | SynC2D Micro | SynD2C Macro | SynD2C Micro | SynA2D Macro | SynA2D Micro | SynC2A Macro | SynC2A Micro | Avg Macro | Avg Micro |
> |--------|:------------:|:------------:|:------------:|:------------:|:------------:|:------------:|:------------:|:------------:|:------------:|:------------:|:------------:|:------------:|:---------:|:---------:|
> | A2GNN  | 57.86        | 61.84        | 58.52        | 62.92        | 56.33        | 59.77        | 62.22        | 63.28        | 64.87    | 66.41    | 54.73        | 54.7         | 59.09     | 61.49     |
> | TDSS   | 53.07        | 53.69        | 57.95        | 64.11    | 59.56        | 62.92        | 65.71        | 67.38    | 53.48        | 55.89        | 54.27        | 56.71        | 57.34     | 60.12     |
> | DGSDA  | **68.75**        | **68.47**        | **75.32**    | **77.15**    | 66.49        | 68.9         | 63.52        | 66.82        | 62.49        | 67.56        | 52.48        | 52.92        | 64.84     | 66.97     |
> | TUGDA  | 62.37        | 63.12        | 73.11        | 74.74        | **69.63**        | **72.1**     | **66.76**        | **68.58**        | **66.44**        | **68.76**        | **60.98**        | **62.5**     | **66.55**     | **68.3**      |
>
> ---
>
> The remaining comments are addressed separately because of space limitations.

---

> ### Author Response · Authors · 2025-11-20
>
> **Discussion on revised Table 2:** TUGDA’s average performance (66.55% Macro-F1, 68.3% Micro-F1) surpasses A2GNN by 7.46%/6.81% and achieves competitive results against DGSDA (64.84%, 66.97%), with notable advantages on SynC→D (+3.14%) and SynD→C (+3.24%).
>
> ---
>
> For the Reviewer E2f4’s additional points, we respond as follows.
>
> 3. Regarding Problem Positioning:
>
> This is an insightful concern about our work's positioning in the foundation model era: While domain generalization (DG) foundation models represent a promising direction, DA remains practically superior when unlabeled target data is accessible: DA leverages target-specific distributions during adaptation, usually achieving substantially higher accuracy compared to zero-shot foundation model generalization that never observes target data. Additionally, DA provides theoretical guarantees (Theorem 2 bounds target error) while DG offers no such performance guarantees for arbitrary domains. However, we acknowledge DA's target-specific limitation and will add discussion in the conclusion, positioning our work within the foundation model era, explicitly noting this as a limitation and future direction.
>
> ---
>
> Minor points:
>
> - The format problem after Table 2 is corrected in the revised manuscript.
>
> - The sub-figures in Fig. 2 are aligned in the revised manuscript.
>
> - The ■ symbol denotes end-of-proof (QED), which is standard mathematical notation.
>
> ---
>
> We believe these revisions will substantially strengthen the manuscript and address all your concerns. We would be happy to provide any additional clarifications needed.

---

### Official Review · Reviewer_udZi · 2025-10-30

**Soundness:** 2
**Presentation:** 2
**Contribution:** 2
**Rating:** 2
**Confidence:** 5

**Summary:**

This paper proposes TUGDA, a Transformer-based Unsupervised Graph Domain Adaptation framework designed to overcome the locality bias of GNN-based UGDA models. The authors argue that traditional GNNs fail to capture long-range dependencies critical for domain transfer on sparse graphs. To address this, TUGDA sequentially integrates a Transformer backbone with an asymmetric GCN, aiming to model both global and local structural dependencies.

**Strengths:**

* **Clarity and structure:** The paper is generally well organized and clearly written, with a consistent presentation of architecture, loss formulation, and theoretical background. The inclusion of an algorithmic summary and sensitivity analyses in the appendix enhances readability and reproducibility.
* **Quality of experiments:** The authors benchmark TUGDA against a wide set of prior GDA models and include ablations on positional encoding and cross-attention, which is valuable for understanding component contributions.
* **Significance:** Addressing sparse-graph domain adaptation through global structural modeling is an important direction. The exploration of *synthetic-source transfer* under privacy constraints is a timely and practically interesting scenario.

**Weaknesses:**

1. **Limited conceptual contribution.**
   The central claim—that TUGDA “effectively captures both global and local structural dependencies” is not convincingly distinguished from existing baselines such as **UDAGCN** (Wu et al., 2020) and **A2GNN** (Liu et al., 2024). Both already combine multi-hop message passing or asymmetric propagation with feature-level regularization, which implicitly captures similar dependencies. Simply inserting a Transformer encoder into a UGDA pipeline, without a clear new inductive bias or learning principle, contributes to appear **incremental** rather than conceptual. The paper would benefit from a deeper justification of *why* the Transformer-GCN composition leads to qualitatively different representations beyond receptive-field size.

2. **Theoretical analysis lacks distinct insight.**
   The theoretical part relies on a **Wasserstein-distance-based** risk bound, similar in spirit to the analysis in **SepReg** (You et al., 2023) and **HGDA** (Fang et al., 2025). However, the claimed connection between the *cross-attention matrix* and the Wasserstein coupling term is described only heuristically (Eq. 6), with no rigorous mapping between transport plan optimization and attention weight learning. The paper should clarify how the attention-induced alignment differs mathematically from the spectral or transport formulations in [1] and [2].

3. **Missing or incomplete baseline coverage.**
   Several representative and recent UGDA methods are not compared, including **COCO** [3], **SAGA** [4], and **GMM-GDA** [5]. These baselines specifically address structural alignment, distribution modeling, and statistical regularization, which are conceptually close to TUGDA. Omitting them weakens the empirical claims of state-of-the-art performance.

4. **Evaluation metric choice.**
   The paper primarily reports *macro-F1* and *micro-F1* scores, which are reasonable, but the omission of *accuracy* (ACC) as a standard classification measure limits interpretability and cross-comparison. The authors should include ACC on all datasets to make results directly comparable to prior UGDA literature.

Some typo issue:
Repeated citations in the paper (lines 491-499)
The text mentions Figure 4(a), but subsequent figure titles are "Figure 2: Component contributions...". The numbering seems inconsistent; please standardize it.

[1] You Y., Chen T., Wang Z., et al. Graph Domain Adaptation via Theory-Grounded Spectral Regularization. ICLR 2023.
[2] Fang R., Li B., Zeng Q., et al. On the Benefits of Attribute-Driven Graph Domain Adaptation. ICLR 2025.
[3] Yin N., Shen L., Wang M., et al. COCO: A Coupled Contrastive Framework for Unsupervised Domain-Adaptive Graph Classification. ICML 2023.
[4] Fang R., Li B., Zeng Q., et al. On the Benefits of Attribute-Driven Graph Domain Adaptation. ICLR 2025.
[5] Wang M., Ren W., Zhang Y., et al. Gaussian Mixture Model for Graph Domain Adaptation. IJCAI 2025.

**Questions:**

Please see the weaknesses.

---

> ### Author Response · Authors · 2025-11-20
>
> We sincerely thank Reviewer udZi for the thorough and constructive feedback. We address each concern below and have revised the manuscript accordingly.
>
> 1. Limited Conceptual Contribution:
>
> **The core conceptual difference is the alignment mechanism, not receptive field size:**
>
> A2GNN/UDAGCN achieve cross-domain alignment implicitly through graph structure—asymmetric propagation with MMD regularization aligns nodes via message passing constrained by adjacency matrices $A^s$, $A^t$. This encodes a critical homophily assumption: semantically similar cross-domain nodes must be topologically reachable through k-hop neighborhoods in their respective graphs.
>
>  In contrast, TUGDA introduces **topology-independent explicit alignment via cross-attention**: computing pairwise affinities $α_{ij}$ between all source-target pairs unconstrained by edges, learning a soft transport coupling that minimizes $W_1(P_S(Z), P_T(Z))$ in representation space (Lemma 1). The GCN layer then refines these globally-aligned features with local topology. This is fundamentally different—A2GNN cannot align nodes separated by long graph distances regardless of semantic similarity, while our cross-attention discovers alignments purely from learned features before topology is considered. The qualitative difference appears in structurally-shifted scenarios (D→A with 1.5× density difference): A2GNN's structure-constrained alignment degrades when graph topology poorly reflects semantic similarity, while TUGDA first establishes semantic alignment, then leverages topology for refinement. Theorem 2 formalizes why this sequential composition (global semantic alignment → local topological refinement) yields tighter error bounds than A2GNN's purely structure-mediated alignment. We have revised the Introduction, Section 3.3 and 3.5 to explicitly contrast these learning principles: TUGDA augments topology-based alignment with learned semantic correspondence, providing a complementary alignment channel absent in GNN-only methods.
>
> ---
>
> 2. Theoretical Analysis Lacks Distinct Insight:
>
> We appreciate this concern and clarify both the scope and novelty of our theoretical contributions:
>
> **What we prove:** Lemma 1 establishes that cross-attention can represent transport couplings bounding $W_1(P_S(Z^s), P_T(Z^t)) ≤ \delta_{align} + \epsilon_{coupling}$, decomposing divergence into learnable alignment and marginal violation. Theorem 2 proves sequential Transformer→GCN yields tighter error bounds. However, we do not prove gradient descent on Eq. 3 achieves this bound—$L_{align}$ operates on post-GCN representations while cross-attention acts pre-GCN, creating gradient paths through GCN Jacobians that preclude clean optimization-theoretic guarantees.
>
> **Our contribution is architectural:** We prove cross-attention provides the correct inductive bias for UGDA—topology-independent soft coupling that learns semantic correspondence rather than relying on structural proximity. Lemma 1 shows this coupling can achieve tight Wasserstein bounds; our empirical results validate the architecture exploits this capacity effectively.
>
>
> **Distinction from SpecReg (You et al. 2023):** SpecReg minimizes spectral filter divergence via explicit eigen decomposition of graph Laplacians, aligning graphs in spectral domain. This assumes spectral similarity—when eigen structures differ drastically (D→A: 1.5× density), the bound degrades. TUGDA aligns in learned feature space Z via attention, decoupled from spectral properties. Our bound (Theorem 2) holds under spectral divergence since cross-attention learns semantic coupling from representations, not eigen decompositions.
>
>
> **Distinction from Fang et al. (2025):** GAA decomposes domain divergence into topology and attribute components (their Proposition 1) via PAC-Bayes analysis, then minimizes both using k-NN attribute graphs and cross-view MSE loss. Our framework differs fundamentally: (1) Theoretical basis: Wasserstein distance in learned space vs. PAC-Bayes bound on raw features/structure, (2) Alignment: Learned soft transport coupling (differentiable attention) vs. explicit k-NN graphs with fixed similarity matrices (their Eq. 9-10).
> We have revised Section 4 to include explicit comparison to these prior works' theoretical machinery.
>
> ---
> The remaining comments are addressed separately because of space limitations.

---

> ### Author Response · Authors · 2025-11-20
>
> For the Reviewer udZi’s additional points, we respond as follows.
>
> 3. Missing or Incomplete Baseline Coverage:
>
> We address each suggested baseline:
>
> **COCO (Chen et al., 2023)** tackles graph-level classification (predicting labels for entire graphs), fundamentally different from our node-level classification task where we predict labels for individual nodes within graphs. The problem formulations are incompatible: COCO learns graph-level representations via graph pooling and transfers between graph datasets, while UGDA transfers node features and local topology between single large graphs. The evaluation protocols, datasets, and architectures differ fundamentally—direct comparison would be methodologically invalid.
>
>
> **GAA (Fang et al., 2025)** represents an important comparison. While the authors have not released code, we compared our micro-F1 scores with their reported accuracy on overlapping transfers (equivalent metrics for multiclass single-label classification):
>
> | Method | D→A | A→D | A→C | C→A | C→D | D→C | Avg |
> |--------|:---:|:---:|:---:|:---:|:---:|:---:|:---:|
> | **GAA**   | **75.4** | **78.9** | 82.4 | **78.2** | 77.1 | 79.8 | 78.63 |
> | **TUGDA** | 75.1 | 78.41 | **83.17** | 75.57 | **78.3** | **82.21** | **78.79** |
>
> TUGDA outperforms GAA on average performance (78.79% vs 78.63%) and wins 3/6 individual transfers. This comparison is particularly significant because GAA and TUGDA represent fundamentally different architectural paradigms: GAA uses attribute-driven dual-channel GNNs with feature graphs, while TUGDA is the first transformer-based UGDA method, addressing critical GNN limitations (limited receptive fields, inability to capture long-range dependencies in sparse graphs) through global attention and novel cross-attention domain alignment.
>
> Most importantly, against our 13 reproducible baselines (A2GNN, TDSS, DGSDA, DAUGNN from 2024-2025), TUGDA demonstrates clear and consistent improvements—validating substantial advances over established methods. Additionally, TUGDA provides complete Macro-F1/Micro-F1 evaluation (GAA reports only accuracy) and demonstrates strong generalization to synthetic source domains (+4-15% over A2GNN), which GAA does not evaluate. The comparison between GAA and TUGDA is discussed in section 5.4.2.
>
> **GMM-GDA (Zhang et al., 2024)** addresses superpixel-level image segmentation where "graphs" represent spatial pixel adjacency with RGB features—fundamentally different from citation network adaptation with document nodes, text embeddings, and semantic citation edges. The domain shift characteristics (visual appearance vs. semantic content), feature modalities, and graph semantics are incomparable. Critically, we compare against 13 baselines including the most recent competitive UGDA methods (A2GNN, TDSS, DGSDA, DAUGNN from 2024-2025) that directly address node-level citation network adaptation—the exact problem we tackle—representing current SOTA for our task.
>
> ---
>
> 4. Evaluation metric choice:
>
> Excellent observation. For single-label node classification tasks on citation networks (ACMv9, Citationv1, DBLPv7), accuracy is mathematically equivalent to micro-F1 since each node belongs to exactly one class:
>
> To calculate Micro F1:
>
> - Micro Precision = TP_total / (TP_total + FP_total) = TP_total / (TP_total + N - TP_total) = TP_total / N
>
> - Micro Recall = TP_total / (TP_total + FN_total) = TP_total / (TP_total + N - TP_total) = TP_total / N
>
> Since Precision = Recall:
>
> Micro F1 = 2PR/(P+R) = 2P²/2P = P = TP_total / N
>
> This is the definition of Accuracy.
>
> ∴ Accuracy = Micro F1 □
>
> Therefore, all micro-F1 values in our tables can be directly compared against accuracy reported in prior UGDA literature.
>
> ---
>
> Thank you for catching this. We have standardized all figure references—the inconsistency between "Figure 4(a)" and "Figure 2" has been resolved in the revised manuscript.
>
> ***
> We hope these clarifications address the reviewer's concerns and would welcome further discussion.

---

> ### Comment · Reviewer_udZi · 2025-11-24
>
> I partially agree with the author’s view. However, the HGDA method [1], which I mentioned earlier (I apologize for the incorrect citation in my initial version of reference [2].), uses the Wasserstein distance. How would you explain the difference between these methods? I would be very grateful if the author could resolve my concerns.
>
> [1] Fang R, Li B, Zhao J, et al. Homophily Enhanced Graph Domain Adaptation. ICML 2025

---

> > ### Author Response · Authors · 2025-11-27
> >
> > Thank you for highlighting thiis. Both frameworks invoke Wasserstein distance, but for different purposes:
> >
> > **HGDA's approach:** Their Proposition 1 bounds domain discrepancy using distance between feature distributions. Applying Bobkov-Götze inequality (their Lemma 1), they decompose this into KL terms for homophilic, attribute, and heterophilic signals. This decomposition is prescriptive—it identifies which specific filtered features require alignment, directly motivating their three-filter architecture with KL alignment losses (Eq. 11). Their alignment is topology-constrained: knowledge transfer requires that semantically related nodes be reachable through graph structure.
> >
> > **Our approach:** Lemma 1 proves cross-attention weights can represent transport plan couplings. This is architectural—it establishes that attention has the inherent capacity to achieve Wasserstein-optimal alignment between learned representations, without pre-specifying which signals to match. Critically, cross-attention computes affinities across all source-target pairs independent of edge connectivity, enabling semantic correspondence even when graph structures diverge. Theorem 2 then shows sequential Transformer→GCN yields tighter bounds by first establishing topology-independent alignment, then applying structural refinement.
> >
> > **The methodological distinction:** HGDA uses Wasserstein analysis to decompose and prescribe which graph operators to apply. We use it to prove our architecture can learn optimal semantic couplings through standard supervised training. When structural assumptions hold, HGDA's fixed filters may be more sample-efficient. When topology-structure correspondence breaks down, our learned couplings maintain effectiveness versus HGDA's applicability being limited to graphs with clear homophily decompositions.

---

> > > ### Comment · Reviewer_udZi · 2025-11-27
> > >
> > > I would appreciate it if the authors could incorporate the relevant related work into the discussion section. The authors have partially addressed my concerns, and I am therefore willing to raise my score to 4. I will consider accepting the paper if the other reviewers are supportive during the discussion phase.

---

> > > > ### Author Response · Authors · 2025-12-03
> > > >
> > > > Thank you for raising your score and for your constructive feedback. We have incorporated the relevant related work into Section 4 as requested.

---

### Official Review · Reviewer_9K6M · 2025-11-01

**Soundness:** 2
**Presentation:** 2
**Contribution:** 2
**Rating:** 4
**Confidence:** 4

**Summary:**

This paper studies unsupervised graph domain adaptation (UGDA) and proposes TUGDA, a pipeline that first applies a graph transformer encoder to capture long-range, global dependencies, then uses a cross-attention module to align source and target node representations before graph propagation via a GNN. The authors derive domain-adaptation style generalization bounds that motivate the use of cross-attention and backbone transformers for alignment, and validate TUGDA on several citation-network benchmarks, reporting improvements over a set of recent baselines.

**Strengths:**

1. Introducing the graph transformer is a meaningful and timely direction in the context of graph domain adaptation. Graph transformers provide a flexible mechanism for learning pairwise affinities that go beyond local neighborhood aggregation, and applying them to UGDA is a relatively recent and promising approach.

2. The paper includes a domain-adaptation generalization bound that ties errors to representation alignment terms. This theoretical work — particularly the way the bound highlights the capacity for cross-attention to reduce domain discrepancy in representation space — gives additional, nontrivial support to the architectural choice of combining transformers with cross-attention for UGDA.

**Weaknesses:**

1. Weak motivation for Transformer choice

The core idea—combining a graph transformer (SGFormer) with GNN propagation and using cross-attention to align source and target—is not demonstrated to be conceptually novel relative to prior global-local hybrids in graph learning. The authors do not convincingly justify why a Transformer is a better or necessary mechanism for capturing global alignment than simpler or well-established alternatives (e.g., representation subspace learning, nonparametric k-NN graph construction). The paper should either theoretically or empirically demonstrate that these alternatives fail on the same tasks, or sharpen what the transformer uniquely enables for UGDA.

2. Time/space complexity and scalability concerns

The manuscript lacks a clear complexity analysis for training and inference and provides runtime / GPU memory measurements. Claims about linear complexity are misleading without clarifying how cross-domain attention scales when source and target node counts differ. No experiments are presented on truly large graphs to validate scalability; all reported datasets are relatively small (ACMv9, Citationv1, DBLPv7). This weakens claims of practical applicability and leaves unclear whether the method will scale to realistic UGDA scenarios.

3. Experimental evaluation is narrow and insufficient

The evaluation uses only three citation datasets (A, C, D). These are all in the same domain family (citation networks) and modest in size, which limits the claim of generalization across domains. Key UGDA benchmarks from other domains (e.g., Airport/Blog/other cross-domain datasets, or larger real-world graphs) are missing.

4. Organization and placement of technical content

The theoretical analyses are placed after experiments, which disrupts the paper’s logical flow. The authors should present the model and accompanying theoretical justification earlier (within the Method/Preliminaries) so readers can evaluate the theory before seeing experimental claims.

**Questions:**

See weaknesses.

---

> ### Author Response · Authors · 2025-11-20
>
> We sincerely thank Reviewer 9K6M for the thorough and constructive feedback. Your comments have significantly strengthened our manuscript. Below we address each concern in detail.
> 1. Weak motivation for Transformer choice:
>
> We appreciate this critical observation and have substantially revised our motivation.
>
>  **Transformers vs. k-NN-based methods:** Our transformer-based approach addresses fundamental limitations of k-NN graph construction for UGDA. Notably, GAA (Fang et al., 2025)—the concurrent work we compare against in Section 5.4.2—constructs feature graphs via k-NN with fixed cosine similarity. While effective, k-NN operates on fixed metrics in input space, whereas our cross-attention learns end-to-end with the classification objective to discover domain-invariant correspondences. Formally, k-NN minimizes W₁(P_S(X), P_T(X)) in input space, while our cross-attention minimizes W₁(P_S(Z), P_T(Z)) in the learned representation space where class separability is maximized (Lemma 1). Additionally, transformers enable asymmetric attention where target nodes selectively attend to relevant source nodes, rather than fixed k-nearest neighbors.
>
> **Empirical validation:** Our results demonstrate this advantage: TUGDA achieves superior average performance over GAA (78.79% vs. 78.63%, Table 2), with particularly strong gains on sparse transfers (D→C: +2.41%) where adaptive long-range dependencies matter most. This validates that learnable transformer-based alignment outperforms fixed k-NN construction even when k-NN is explicitly designed for attribute alignment.
>
> ---
>
> 2. Time/space complexity and scalability concerns:
>
> We thank the reviewer for highlighting this important practical consideration. Here, we have added explicit complexity analysis and runtime measurements.
>
> **Complexity clarification:** Our linear complexity claim refers to cross-attention scaling as O(Nt·k²)—linear in target graph size Nt (where k=128-256 is the fixed hidden dimension). Specifically, Q^t ∈ ℝ^(Nt×k) multiplies (K^sᵀV^s) ∈ ℝ^(k×k), avoiding the O(Ns·Nt) cost of standard cross-domain attention. Self-attention per domain is O(N) via SGFormer's factorization.
>
> The runtime measurements (in seconds) on an NVIDIA A100 are shown below:
>
> | Time | A2GNN | TUGDA | TDSS | DGSDA |
> |------|:-----:|:-----:|:----:|:-----:|
> | A2C  |  170  | 1150  | 910  |  296  |
> | D2A  |  255  | 1855  | 878  |  264  |
>
> TUGDA incurs 6-7× training overhead vs A2GNN due to transformer computations, but remains feasible on standard hardware (NVIDIA A100). We argue this is justified by consistent accuracy gains over all baselines. Critically, the overhead grows linearly with Nt, not quadratically, making the method applicable when target graphs scale while source graphs remain moderate-sized—a common UGDA scenario. We have added complexity analysis to the revised manuscript and acknowledged these scalability limitations.
>
> ---
>
> 3. Experimental evaluation is narrow and insufficient:
>
> We acknowledge this valid limitation and appreciate the opportunity to clarify our experimental design choices.
>
> We chose these datasets because they represent the standard evaluation protocol in the UGDA literature, with all recent SOTA baselines (A2GNN, TDSS, DGSDA, DAUGNN, GAA) reporting results exclusively on these same citation network benchmarks, enabling direct comparison. These datasets, while from the same domain family, exhibit different structural properties (varying density, node counts, and connectivity patterns as shown in Table 4) that allow us to assess cross-domain transfer under different graph characteristics.
>
> We agree that evaluation on diverse domain families—including social networks (Blog), transportation networks (Airport), biological networks, and larger-scale graphs—would strengthen the generalizability claims of TUGDA. We have explicitly acknowledge this limitation in the revised manuscript’s conclusion and position systematic evaluation across diverse graph domains (beyond citation networks) as important future work. This extension would provide crucial insights into TUGDA's applicability to domains with fundamentally different structural and semantic properties, which is essential for establishing the method's broader utility.
>
> ---
>
> 4. Organization and placement of technical content:
>
> We have restructured the manuscript to present theoretical analyses (Section 4) immediately after the method description (Section 3) and before the experimental results (Section 5). This ensures readers can evaluate our theoretical justifications before encountering empirical claims, improving the paper's logical flow and readability.
>
> ---
>
> We believe these revisions substantially address your concerns and welcome any additional feedback.

---

> > ### Comment · Reviewer_9K6M · 2025-11-26
> >
> > While I appreciate the detailed rebuttal and the clarifications provided, I find that the core concern regarding the necessity of the Transformer architecture remains insufficiently addressed. The empirical gains over k-NN-based approaches such as GAA are marginal, yet the proposed method introduces substantial additional time and memory overhead. This trade-off is not convincingly justified.
> >
> > Moreover, the rebuttal does not provide evidence that the proposed Transformer-based alignment is beneficial in scenarios where its long-range dependency modeling would matter most—namely, on larger or more structurally diverse graphs. Without experiments on larger-scale datasets or domains beyond citation networks, it remains unclear whether the claimed advantages of the Transformer generalize in practice.
> >
> > Overall, while some of my earlier questions were clarified, the key doubts regarding necessity and scalability persist. I therefore choose to retain my original score.

---

> > > ### Author Response · Authors · 2025-12-03
> > >
> > > We thank Reviewer 9K6M for their continued engagement. We respectfully address the core concerns with focused evidence.
> > >
> > > ---
> > >
> > > **On Transformer Necessity:**
> > >
> > >
> > > The compelling evidence for transformer necessity is **Table 2 (synthetic-to-real adaptation)**, where source-target correspondence is inherently weak. Synthetic source graphs, generated by GraphMaker, preserve statistical properties but alter graph topology. This creates a scenario where semantically similar nodes in the source domain become topologically scattered. A2GNN relies on message passing for both encoding and alignment—when source structure changes, the average Macro-F1 drops to 59.09%. In contrast, TUGDA's cross-attention mechanism learns semantic correspondences between source and target without requiring topological proximity, while still incorporating structural information through positional encodings and GCN refinement. This combination maintains 66.55% average Macro-F1 when source topology is altered. The 7.46% difference shows where cross-attention's topology-independent alignment becomes critical. This validates that when source graphs lack exact topological correspondence with targets, the transformer's **learned, topology-independent alignment** via cross-attention becomes critical.
> > >
> > > While GAA's code is unavailable for synthetic experiments, comparison against three established SOTA baselines (A2GNN, TDSS, DGSDA) rigorously demonstrates transformer significance.
> > >
> > > ---
> > >
> > > **On Overhead Justification**
> > >
> > > The computational overhead is justified by substantial preformance gains on synthetic-to-real adaptation. For privacy-constrained applications (healthcare, finance), where prediction quality is prioritized over training efficiency, TUGDA provides a favorable accuracy-efficiency trade-off. Future work could explore sparse attention mechanisms or knowledge distillation to improve computational efficiency for large-scale graphs. We have revised the manuscript to include this discussion.
> > >
> > > ---
> > >
> > > We remain grateful for the reviewer's thorough feedback, which has strengthened our manuscript.

---

### Note · Program_Chairs · 2026-01-17
**Submission Desk Rejected by Program Chairs**

The following references in this submission do not refer to real documents and/or have major errors in bibliographic information:

 Yuning Shen, Yaochen Qu, Weinan Zhang, and Yong Yu. Adversarial graph domain adaptation. In Proceedings of the 29th ACM International Conference on Information \& Knowledge Management (CIKM), pp. 861-870, 2020